# A rationally designed JAZ subtype-selective agonist of jasmonate perception

Yousuke Takaoka[1,2], Mana Iwahashi[1], Andrea Chini [3], Hiroaki Saito[4], Yasuhiro Ishimaru[1], Syusuke Egoshi[1], Nobuki Kato[1], Maho Tanaka[5], Khurram Bashir[5], Motoaki Seki [5], Roberto Solano[3] & Minoru Ueda [1,6]

The phytohormone 7-*iso*-(+)-jasmonoyl-L-isoleucine (JA-Ile) mediates plant defense responses against herbivore and pathogen attack, and thus increases plant resistance against foreign invaders. However, JA-Ile also causes growth inhibition; and therefore JA-Ile is not a practical chemical regulator of plant defense responses. Here, we describe the rational design and synthesis of a small molecule agonist that can upregulate defense-related gene expression and promote pathogen resistance at concentrations that do not cause growth inhibition in *Arabidopsis*. By stabilizing interactions between COI1 and JAZ9 and JAZ10 but no other JAZ isoforms, the agonist leads to formation of JA-Ile co-receptors that selectively activate the JAZ9-EIN3/EIL1-ORA59 signaling pathway. The design of a JA-Ile agonist with high selectivity for specific protein subtypes may help promote the development of chemical regulators that do not cause a tradeoff between growth and defense.

[1] Department of Chemistry, Graduate School of Science, Tohoku University, Sendai 980-8578, Japan. [2] Precursory Research for Embryonic Science and Technology (PREST), Japan Science and Technology Agency, Tokyo 102-0076, Japan. [3] Plant Molecular Genetics Department, National Centre for Biotechnology (CNB), Consejo Superior de Investigaciones Cientificas (CSIC), Campus University Autonoma, 28049 Madrid, Spain. [4] Center for Biosystems Dynamics Research, RIKEN, Suita 565-0874, Japan. [5] Plant Genomic Network Research Team, RIKEN Center for Sustainable Resource Science, Yokohama 230-0045, Japan. [6] Department of Molecular and Chemical Life Sciences, Graduate School of Life Sciences, Tohoku University, Sendai 980-8578, Japan. Correspondence and requests for materials should be addressed to M.U. (email: minoru.ueda.d2@tohoku.ac.jp)

Plant hormones are chemical regulatory factors that function in physiological events throughout a plant's life cycle, such as development, differentiation, reproduction, stress tolerance, and immune responses[1,2]. The receptors of most plant hormones have already been identified[1–3]; some plant hormones induce protein-protein interactions (PPIs) and modulate multiple plant responses in parallel[4]. The plant hormone 7-*iso*-(+)-jasmonoyl-L-isoleucine (JA-Ile, **2**)[5], the active form of jasmonic acid (JA, **1**), plays an important role in plant defense responses against environmental stresses (Fig. 1a)[6,7]. The most important physiological role of **2** is the activation of induced immunity, which is triggered by attack from insect herbivores and necrotrophic pathogens, as well as tissue injury[8]. Thus, upregulating **2**-mediated defense responses is expected to reinforce plant resistance against foreign invaders. JA-Ile (**2**) induces PPI between CORONATINE INSENSITIVE1 (COI1; the F-box subunit of the skp/Cullin/F-box-type ubiquitin ligase complex) and JASMONATE ZIM DOMAIN (JAZ) transcriptional repressor proteins[9–11], leading to plant defense responses, as well as plant growth inhibition or senescence[6,7]. Such a growth-defense trade off[12] is partly due to resource allocation in the plant body, as upregulating defense responses requires plant nutrients, thereby suppressing plant growth[8]. Thus, plant defense responses are activated temporarily at the cost of plant growth only when the plants suffer attack by foreign enemies. This dichotomy hinders the use of **2** as a plant defense regulator, and strategies for uncoupling the plant growth and defense responses triggered by **2** are keenly desired.

The molecular basis of this growth-defense trade-offs has attracted much attention[13]. *COI1* and 13 subtypes of *JAZ* are

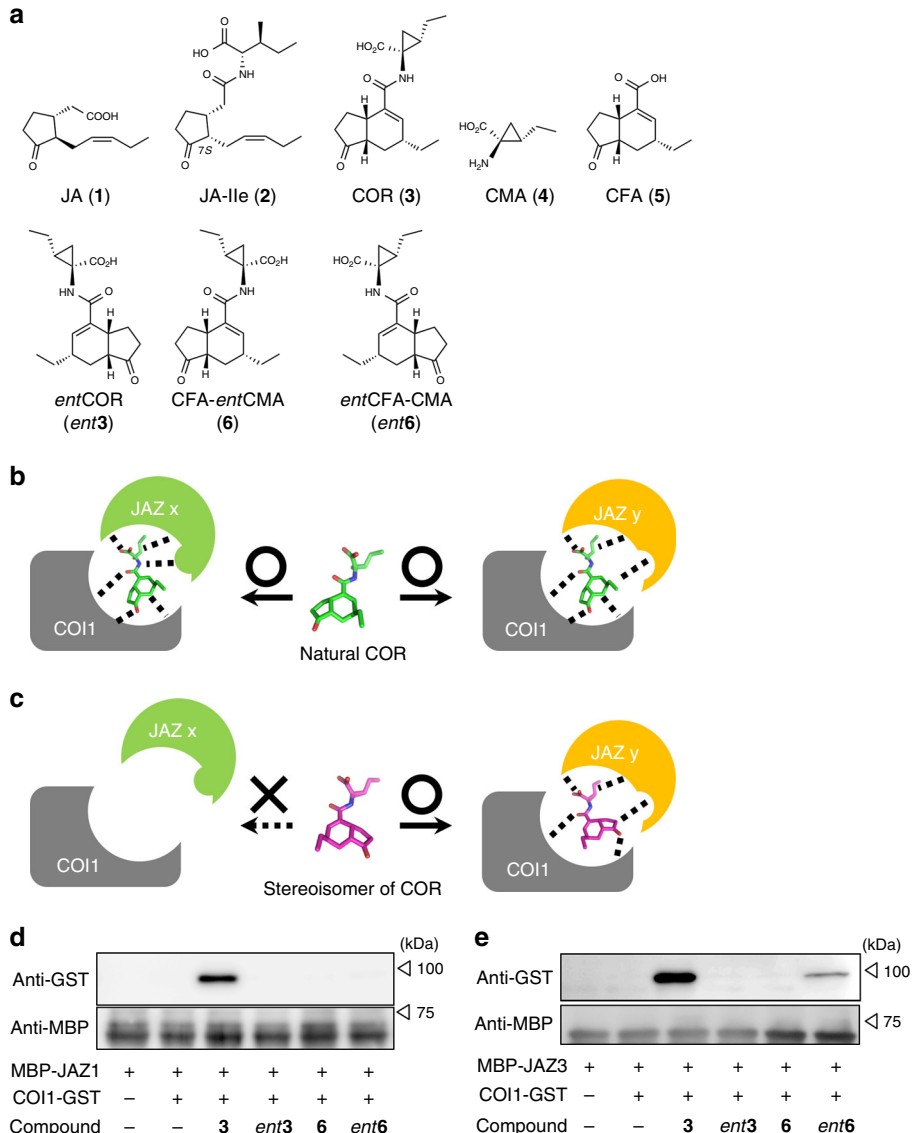

**Fig. 1** One stereoisomer of COR is a potential JAZ subtype-selective agonist. **a** Chemical structures of jasmonate derivatives and coronatine diastereomers. **b, c** Schematic diagram of ligand-induced protein-protein interactions (PPIs) between COI1-JAZ co-receptors; **b** naturally occurring ligands (i.e., coronatine) can interact with all co-receptors, whereas **c** the stereochemical isomer (used in this study) can interact with only some co-receptors. **d, e** Pull down assay of purified GST-COI1 (5 nM) with recombinant proteins expressed in *E. coli*, including **d** MBP-JAZ1 (full length, approximately 40 nM), and **e** MBP-JAZ3 (full length, approximately 40 nM), in the presence of COR derivatives (100 nM). GST-COI1 bound to MBP-JAZ proteins was pulled down with amylose resin and analyzed by immunoblotting. Goat HRP-conjugated anti-GST antibody was used to detect GST-COI1 (black triangles). Rat anti-MBP antibody and goat HRP-conjugated rat-IgG antibody were used to visualize MBP-JAZ protein levels as the input materials (white triangles)

encoded in the genome of *Arabidopsis thaliana*; **2** can induce PPI between COI1 and most of JAZ subtypes[14–17] to cause various physiological responses, as described above. The upregulated JA responses depend on the degradation of the specific subtype of JAZ repressor recruited by the COI1-associated ubiquitin ligase that in turn activates unique subsets of transcription factors[15]. Thus, PPI induction between COI1 and specific JAZ subtypes is only involved in plant defense responses. Thus, it might allow activation of plant defenses but do not cause growth inhibition. However, the detailed physiological functions of all JAZ subtypes remain unclear, as genetic analyses have sometimes provided enigmatic results due to the complexity of the JA-mediated signaling cascade—the genetic redundancy of *JAZ* genes, the involvement of multiple co-acting factors[15], and signaling cross-talk with other phytohormones[18–21] have all been observed; and a well-known antagonistic interaction occurs between JA-mediated defense responses against necrotrophs and salicylic acid (SA)-mediated defense responses against biotrophs[21–23].

Thus, powerful chemical tools for the study of the JA-mediated signaling cascade are strongly desired[24,25]. For example, Monte et al.[26] rationally designed and developed a general antagonist of the COI1-JAZ co-receptor based on a specific modification of the natural product (+)-coronatine (**3**)[27], a structural mimic of **2** (Fig. 1a)[5,28]. In contrast, a JAZ subtype-selective agonist could help to uncouple the plant growth-defense trade-off.

Here, we find that a stereoisomer of **3** functions as a PPI stabilizer with moderate JAZ-subtype selectivity inducing COI1-JAZ co-receptor formation (Fig. 1c). Additionally, based on the stereoisomer **3**, we succeed in the rational design of a JAZ subtype-selective agonist that induces PPI between only two JAZ proteins and COI1. This is achieved by combining the results of an in silico docking study with those of novel in vitro assay systems to evaluate PPIs between COI1 and all JAZ subtypes. Furthermore, detailed assessment of *Arabidopsis* phenotypes combined with the gene expression analyses and fungal infection assays reveal that our JAZ subtype-selective agonist enhances defense responses in *Arabidopsis* against pathogenic infection at concentrations that do not cause growth inhibition. The unique bioactivity of our JAZ subtype-selective agonist can be attributed to the uncoupling of the growth-defense trade-off through the selective activation of JAZ9-EIN3/EIL1- ERF1/ORA59 signaling.

## Results

**A stereoisomer of 3 is a potential JAZ-selective agonist**. Naturally occurring **3** can induce PPIs between COI1 and most of the JAZ subtypes[14,15,17], suggesting that it bears the structural features and/or overall 3D shape necessary to interact with them[17]. Thus, analogs of **3** with partially altered shapes compared with the original molecule might exhibit selectivity for PPI induction between COI1 and some JAZ subtypes over others (Fig. 1b,c). We therefore used a stereochemical isomer of **3** to develop a subtype-selective PPI stabilizer between COI1 and JAZ. Optically pure samples of the building blocks of **3**, (+)-coronamic acid (CMA, **4**)[29], and (+)-coronafacic acid (CFA, **5**)[30], were coupled to give naturally occurring **3**, the enantiomer *ent***3**, and the stereochemical hybrid isomers, CFA-*ent*CMA (**6**) and *ent*CFA-CMA (*ent***6**), respectively (Fig. 1a)[29].

Pull-down experiments using recombinant maltose-binding protein-tagged JAZ1 (MBP-JAZ1)[10] and glutathione-*S*-transferase-tagged COI1 (GST-COI1)[17] demonstrated that only **3** caused PPI between GST-COI1 and MBP-JAZ1 (Fig. 1d), whereas three other isomers were inactive. In contrast, and to our surprise, both *ent***6** and **3** caused PPI between GST-COI1 and MBP-JAZ3 (Fig. 1e), suggesting that *ent***6** might function as an agonist with

JAZ subtype-selectivity causing PPIs between COI1 and some of JAZ subtypes, including JAZ3.

**In vitro binding assay systems with JAZ short peptide**. To examine the JAZ-subtype selectivity of *ent***6**, we developed a versatile assay system for COI1/JAZ PPI detection. According to a report on the crystal structure of the COI1/**2or3**/JAZ co-receptor complex[17], short (27 amino acids) peptide fragments composed of Jas motifs of a JAZ protein are sufficient for co-receptor formation. Based on this finding, we developed an in vitro binary-tag pull-down system for PPI detection that covers PPIs between COI1 and all JAZ subtypes. The requisite short peptides were easily prepared via Fmoc-based solid phase peptide synthesis[17]. The Jas motifs of JAZ1 and 2 are almost identical and those of JAZ5 and 6 are identical (Supplementary Fig. 1a). JAZ7, 8, and 13 are expected to have little if any affinity with COI1 because of the lack of critical COI1 binding sequence (RR or RK) in their Jas motifs (Supplementary Fig. 1a)[31,32]. Thus, we prepared nine short peptides of 13 JAZ subtypes including JAZ1/2, JAZ3-6, JAZ9-12 (Supplementary Fig. 24). JAZ13 was also prepared as a negative control. Since only a few of them contain a Cys in their sequences (Supplementary Fig. 1a), we added Cys at their *N*-termini in order to allow their labeling with Oregon green® (OG) as an epitope tag for pull-down purification (Fig. 2a, b and Supplementary Figs. 1, 2)[33]. For the Cys-containing JAZ13 (Supplementary Fig. 1a), 5-carboxy-OG was introduced in the *N*-terminus of JAZ13 peptide. Each subtype of OG-conjugated JAZ short peptides (OG-JAZ) was mixed with GST-COI1 and **3**. As OG and GST functions as binary tags in the ternary complex, the resulting complex can be pulled down by an anti-fluorescein antibody and detected with HRP-conjugated anti-GST antibody (Fig. 2a). This binary-tag system for PPI detection worked well, as **3** strongly induced PPIs between COI1 and OG-JAZ subtypes except for OG-JAZ13 used as a negative control (Fig. 2c, d). Synthetic **2** containing (3 *R*, 7 *R*) and (3 *R*, 7 *S*) isomers in a 95:5 ratio was also weakly effective for PPI induction using OG-JAZ1, since the minor isomer (3 *R*, 7 *S*)-**2** is a strong agonist of the COI1-JAZ co-receptor (Fig. 2e)[5]. In contrast, these PPIs were not observed for *ent***3** and **1**. These results are broadly consistent with previous reports[5,16], which validates the reliability of our binary-tag pulldown assay. In addition, PPI between COI1 and full-length JAZ1 was competitively inhibited by OG-JAZ1 in a dose-dependent manner (Supplementary Fig. 3). Next, we used this binary-tag system to evaluate the selectivity of *ent***6** for different JAZ subtypes. As shown in Fig. 2c, among all OG-JAZ subtypes, *ent***6** strongly induced PPI for OG-JAZ3, OG-JAZ11, and OG-JAZ12, and weakly for OG-JAZ9 and OG-JAZ10, whereas **6** failed to induce PPI. All of these PPIs depended on the concentration of *ent***6** used (Supplementary Fig. 4a). In contrast, *ent***6** caused PPI for OG-JAZ1, 4 and 5/6 but only at concentrations of *ent***6** as high as 3000 nM, a concentration over 100-fold higher than required for OG-JAZ3, 9, 10, 11, and 12 (Supplementary Fig. 4b), suggesting the significantly lower affinity of *ent***6** for these co-receptors. Little correlation was observed between the sequence homology of the JAZ subtypes and their individual affinity for *ent***6** (Supplementary Fig. 4c). All of these results demonstrate that we succeeded in tuning the JAZ subtype selectivity of **3** using the stereoisomer *ent***6**.

**Rational design of a subtype-selective agonist**. As shown in the previous section, the JAZ-subtype selectivity of *ent***6** was insufficient for practical use. We therefore tried to optimize it using in silico docking and molecular dynamics (MD) analyses. Model structures of JAZ3, 9, 10, 11, and 12 were constructed based on the crystal structure of COI1-**3**-JAZ1 complex using Swiss-

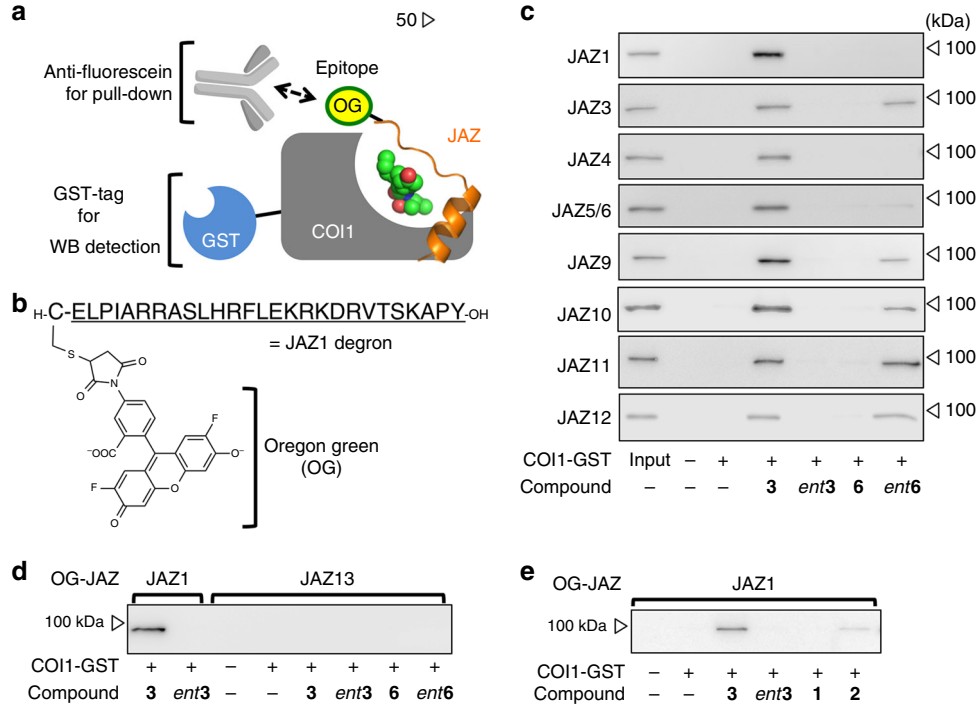

**Fig. 2** In vitro COI1-JAZs binding assay with epitope-conjugated JAZ short peptide. **a** Schematic of in vitro co-immunoprecipitation assay system for agonists of COI1-JAZ co-receptors using epitope-conjugated JAZ degron peptide and GST-COI1. An anti-fluorescein antibody was used to purify the ternary complex, and a GST-tag was used for the immunoblot assay with an anti-GST antibody. **b** Molecular design of the Oregon green (OG)-conjugated Jas motif of JAZ1. **c** Pull-down assay of purified GST-COI1 (5 nM) with each OG-conjugated JAZ peptide (10 nM) in the presence of stereoisomers of (+)-COR (**3**, *ent***3**, **6**, or *ent***6**, 100 nM). **d** Pull-down assay of purified GST-COI1 (5 nM) with OG-conjugated JAZ13 peptide (10 nM) in the presence of **3**, *ent***3**, **6**, or *ent***6** (100 nM). **e** Pull-down assay of purified GST-COI1 (5 nM) with OG-conjugated JAZ1 peptide (10 nM) in the presence of **3**, *ent***3**, **1** or **2** (100 nM); Goat HRP-conjugated anti-GST antibody was used to detect GST-COI1 in **c**–**e**

PdbViewer. The **3** in each modeled complex was replaced with *ent***6** via docking simulations. The models of these complexes were then used for subsequent MD simulations in water. MD simulations were carried out for 500 ns to investigate the differences of hydrogen bond networks between the compounds and surrounding residues in the binding pocket of each system. In these MD simulations, the COI1-**3**-JAZ1 system was used as a reference, and the results of analysis were compared with those of the COI1-*ent***6**-JAZ systems (JAZ3, 9, 10, 11, and 12). Figure 3a and Supplementary Fig. 5a show the typical bound structure of **3** obtained from the MD simulation of COI1-**3**-JAZ1 and the radial distribution functions (RDF) curves for the possible hydrogen bond pairs. These results showed that the ketone oxygen of **3** mainly forms hydrogen bonds with the NH1-proton of R496 (COI1) and NH-proton of A204 (JAZ1) during the MD simulation, indicating the importance of these hydrogen bonds for the binding of JAZ1 with COI1. On the other hand, in cases of the COI1-*ent***6**-JAZ systems, the formation of hydrogen bonds with the ketone oxygen of *ent***6** depend on the JAZ-subtypes (see Fig. 3b, c and Supplementary Fig. 5b-e). For instance, in the case of COI1-*ent***6**-JAZ9, while the ketone oxygen of *ent***6** formed a hydrogen bond with the NH1-proton of R496 (COI1), a less frequency of hydrogen bond formation was observed with the NH1-proton of A222 (JAZ9) (Fig. 3b and Supplementary Fig. 5e). These results suggest an additional unoccupied space around the ketone group of *ent***6** would arise when complexed with COI1 and a part of JAZ subtypes, and thus the JAZ-subtype selectivity of *ent***6**-derivatives might be altered by structural modifications of the ketone group.

Based on the results of the docking and MD studies, we designed and prepared three oxime derivatives of *ent***6**, all stable

in plant cultured medium, and measured their JAZ-subtype selectivity (**7**–**9**, Fig. 3d and Supplementary Figs. 25–30). Our binary-tag pulldown system for PPI revealed that the JAZ-subtype selectivity of **7** was nearly equal to that of *ent***6**, whereas **9** was ineffective in this assay (Fig. 3e). In contrast, high JAZ-subtype selectivity was observed for *O*-phenyl oxime **8**, which selectively induced PPI between COI1 and OG-JAZ9 or OG-JAZ10 (Fig. 3e and Supplementary Fig. 6). The in silico docking simulations also showed that **8** can bind to the same binding pocket of COI1/JAZ9 and 10 (Supplementary Fig. 7). The obtained binding poses were similar to that of **3** in the COI1-**3**-JAZ1 complex and were kept during the subsequent long-term MD simulation. We further confirmed the JAZ-selectivity of *ent***6** or **8** in an in planta assay using transgenic β-glucuronidase (GUS)-reporter *Arabidopsis* lines, including *35 S:JAZ1-GUS*, *35 S:JAZ9-GUS*, *35 S:JAZ10-GUS*, *35 S:JAZ11-GUS*, and *35 S:JAZ12-GUS* (Fig. 3f, g, and Supplementary Fig. 8)[26]. The degradation of the JAZ-GUS fusion protein was visualized as reduced GUS staining in all of five GUS-reporter *Arabidopsis* lines treated with **3**. In contrast, *ent***6** and **8** triggered the degradation of JAZ-GUS in the JAZ9-GUS and JAZ10-GUS-reporter lines, whereas almost no degradation was observed in the JAZ1-GUS, JAZ11-GUS, and JAZ12-GUS line. This confirmed that **8** functions as a JAZ9/10-selective PPI-stabilizer both in vitro and in vivo.

**Selective activation of JA responses by designed agonist.** We next individually examined the effects of **8**, *ent***6**, and **3** on physiological responses and gene expression in *Arabidopsis* (Col-0) seedlings. Growth inhibition and anthocyanin accumulation are well-known responses induced by jasmonates, including **3**[34,35], and it was hoped that these effects would not be observed for **8**.

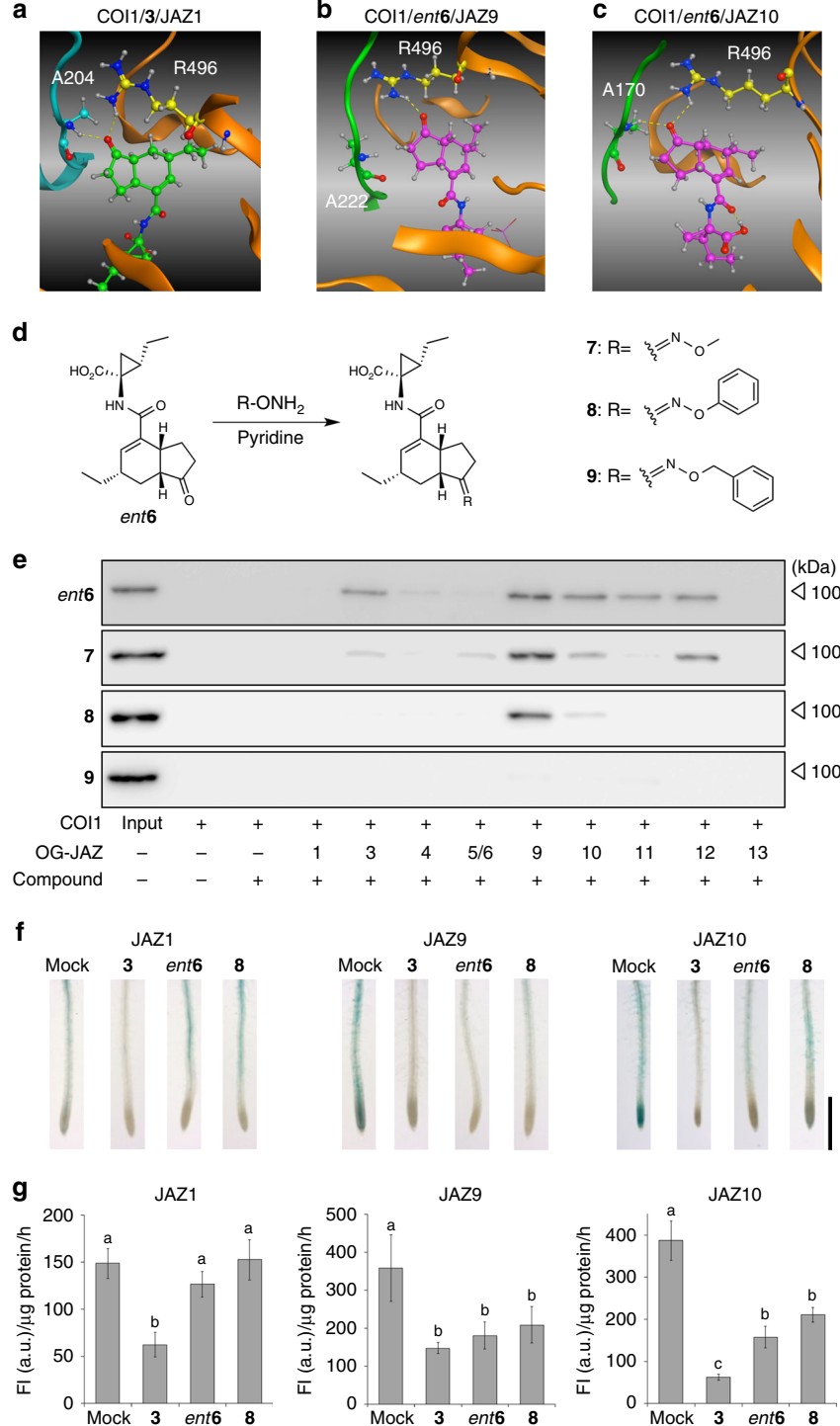

**Fig. 3** Rational design of JAZ subtype-selective agonist by an in silico docking study. **a** The obtained average structure of MD simulation of COI1-**3**-JAZ1. The hydrogen bond between the ketone group of **3** with A204$^{JAZ1}$ or R496$^{COI1}$ is indicated by a yellow dotted line. **b** The obtained average structure of COI1/JAZ9 complexed with *ent***6**, which was constructed by in silico docking analyses and MD simulation. The hydrogen bond between the ketone group of *ent***6** with R496$^{COI1}$ is shown as yellow dotted line. **c** The obtained average structure of COI1/JAZ10 complexed with *ent***6**, which was constructed by in silico docking analyses and MD simulation. The hydrogen bond between the ketone group of *ent***6** with A170$^{JAZ10}$ (corresponding to A204$^{JAZ1}$) or R496$^{COI1}$ is indicated by a yellow dotted line. **d** Synthesis scheme for compounds **7**–**9** from *ent***6** as a starting material and corresponding oxime molecules. **e** Pull-down assay of purified GST-COI1 (5 nM) with all OG-conjugated JAZ peptides (10 nM) in the presence of *ent***6**, **7**, **8**, or **9** (500 nM). HRP-conjugated anti-GST antibody was used to detect GST-COI1. **f** Evaluation of GUS activity in the roots of 4-day-old 35S:JAZ1-GUS, 35S:JAZ9-GUS, and 35S:JAZ10-GUS plants. Seedlings were pretreated for 30 min with or without ligand (**3**, *ent***6**, or **8**, 1 μM), and stained with 5-bromo-4-chloro-3-indolyl glucuronide; the experiments were repeated three times with similar results. Scale bar, 1 mm. **g** Quantification of GUS activity in 20 roots of 4-d-old 35S:JAZ1-GUS, 35S:JAZ9-GUS, and 35S:JAZ10-GUS plants ($n = 4$). Significant differences were evaluated by one-way ANOVA/Tukey HSD post hoc test ($p < 0.01$). Seedlings were pretreated as described above. Three independent replicates were measured, and values represent mean ± s.d. (Supplementary Methods)

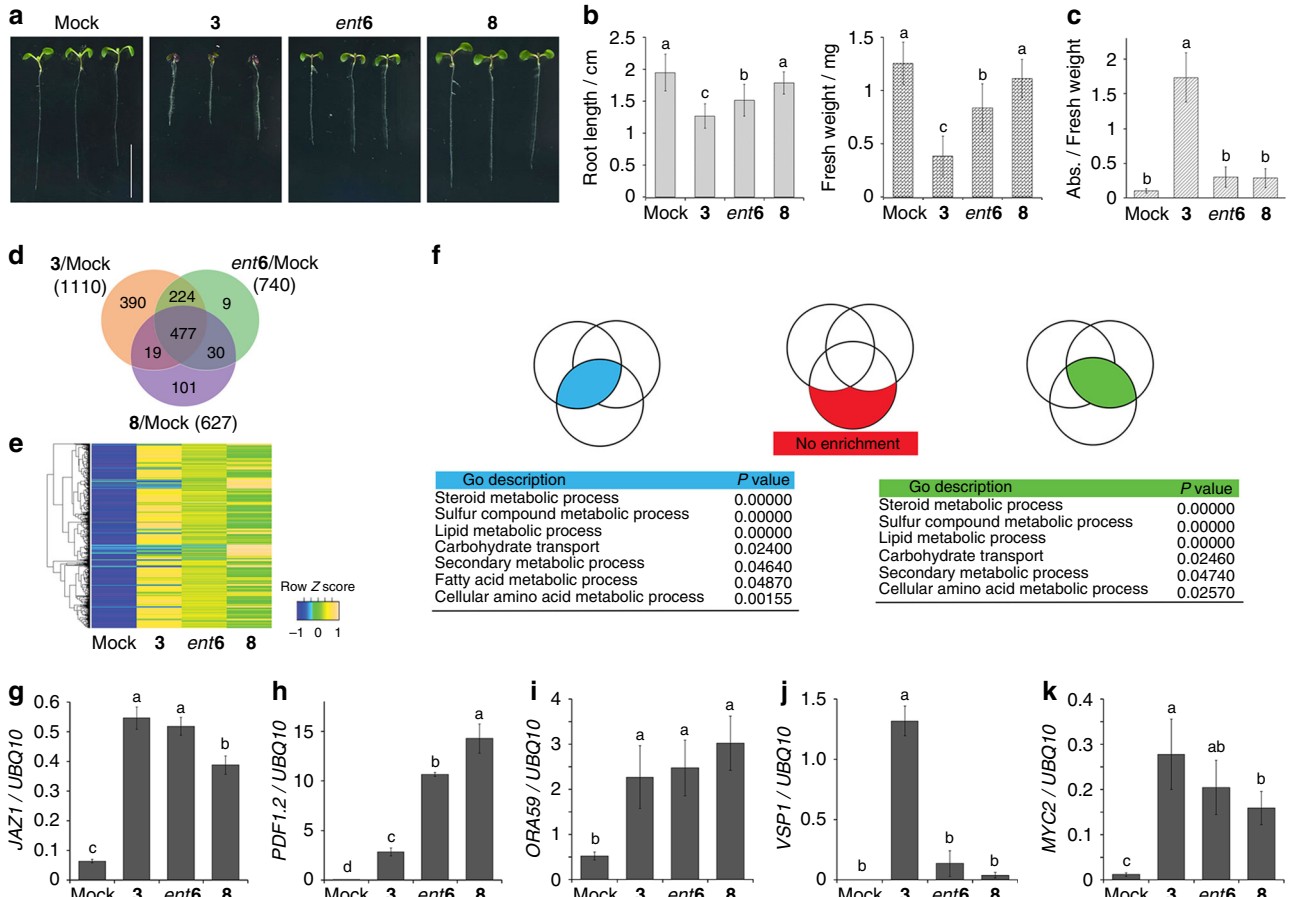

**Fig. 4** JA responses and microarray analyses in **3**/*ent***6**/**8**-treated *Arabidopsis* seedlings. **a** WT *Arabidopsis* seedlings grown for 6 days on 1/2 MS medium containing **3**, *ent***6**, or **8** (1 μM). Scale bar, 10 mm. **b** Quantification of root length or fresh weight of the aerial part in the ligand-treated seedlings shown in **a** ($n = 18$). Significant differences were evaluated by one-way ANOVA/Tukey HSD post hoc test ($p < 0.01$). **c** Quantification of accumulated anthocyanins in the ligand-treated seedlings shown in **a**. ($n = 15$). Significant differences were evaluated by one-way ANOVA/Tukey HSD post hoc test ($p < 0.01$). **d** Venn diagram indicating the number of genes up-regulated at least 2.5 times, in response to different treatments ($p < 0.05$, FDR followed by Tukey's HSD as post hoc test). **e** Heat map illustrating changes in gene expression in response to different treatments. Genes with at least a 2.5-fold increase in expression by **8** ($p < 0.05$, FDR followed by Tukey's HSD as post hoc test). **f** Go enrichment analysis of Fig. 4d. **g–k** Analysis of JA-responsive gene expression by quantitative RT-PCR (qRT-PCR) in 6-d-old WT *Arabidopsis* seedlings with or without ligands (**3**, *ent***6**, or **8**, 1 μM) treatment for 2 h (**g**, **i**, **k**) or for 8 h (**h**, **j**) ($n = 3$). Significant differences were evaluated by one-way ANOVA/Tukey HSD post hoc test ($p < 0.01$)

Growth analyses were undertaken using plates of ligand-containing agar or by repeatedly dropping the ligand solution on the leaf. It was found that the growth of both the root and aerial parts of *Arabidopsis* were strongly inhibited by **3**; less so by *ent***6**; and hardly at all by **8** (Fig. 4a, b and Supplementary Figs. 9a, b). Almost no growth inhibition was observed with the repetitive treatment of both *ent***6** and **8** in the aerial part of *Arabidopsis* (Supplementary Fig. 9c, d). Moreover, **3** strongly induced anthocyanin or glucosinolate accumulation as previously reported[5,15], whereas *ent***6** or **8** did not (Fig. 4c and Supplementary Fig. 9e). To examine the mode of action (MOA) of **8**, DNA microarray analyses were carried out for comprehensive analyses of gene expression. When WT plants were treated with **8**, 627 genes were induced, 477 of which were also induced both by **3** and *ent***6** (Fig. 4d, e, Supplementary Fig. 10a, b and Supplementary Data 1). GO enrichment analysis showed that **8** does not have any significant off target categories (Fig. 4f and Supplementary Data 2). Although several *JAZs* marker genes for early JA responses controlled by the COI1-JAZ co-receptor were included among these up-regulated genes, induction of these genes by **8** was lower than those by **3** and *ent***6** (Fig. 4g and Supplementary Data 1). Surprisingly, **8** strongly activated the

expression of *PDF1.2*[36], a marker gene for defense responses against infection by necrotrophic pathogens, with an expression level at least 4-fold higher than that induced by **3** (Fig. 4h; Supplementary Data 1 and Supplementary Fig. 11a). Additionally, *ORA59*[37], which encodes a transcription factor that directly regulates *PDF1.2*[38–40], was also more activated by **8** than by either **3** or *ent***6** (Fig. 4i and Supplementary Fig. 11b). Moreover, the expression of *ERF1* and other *ORA59*/*ERF*-regulated defense response genes, such as *HEL* or *B-chi*, were also activated by **8**, as well as **3** and *ent***6** (Supplementary Fig. 12a). Intriguingly, **8** failed to upregulate *VSP1*[35,41], a marker gene for wounding-induced defense responses, but moderately upregulated the transcription factor gene *MYC2*[42,43] (Fig. 4j, k and Supplementary Fig. 11c, d). Similarly, **8** slightly upregulated **1**-biosynthetic genes such as *AOS*, *OPR3*, and *LOX3* (among early-acting JA-responsive genes), or *LOX2* (late-acting JA-responsive gene) whereas both *ent***6** and **3** strongly upregulated these genes (Supplementary Fig. 12b, c). The induction of these genes by **8** would not be attributed to the presence of endogenous **2** because the same gene expression patterns for *PDF1.2*, *ORA59*, *VSP1*, and *MYC2* were also observed in the *jar1* mutant[44–46] (in which the biosynthesis of **2** decreased) treated with *ent***6** or **8** (Supplementary Fig. 13).

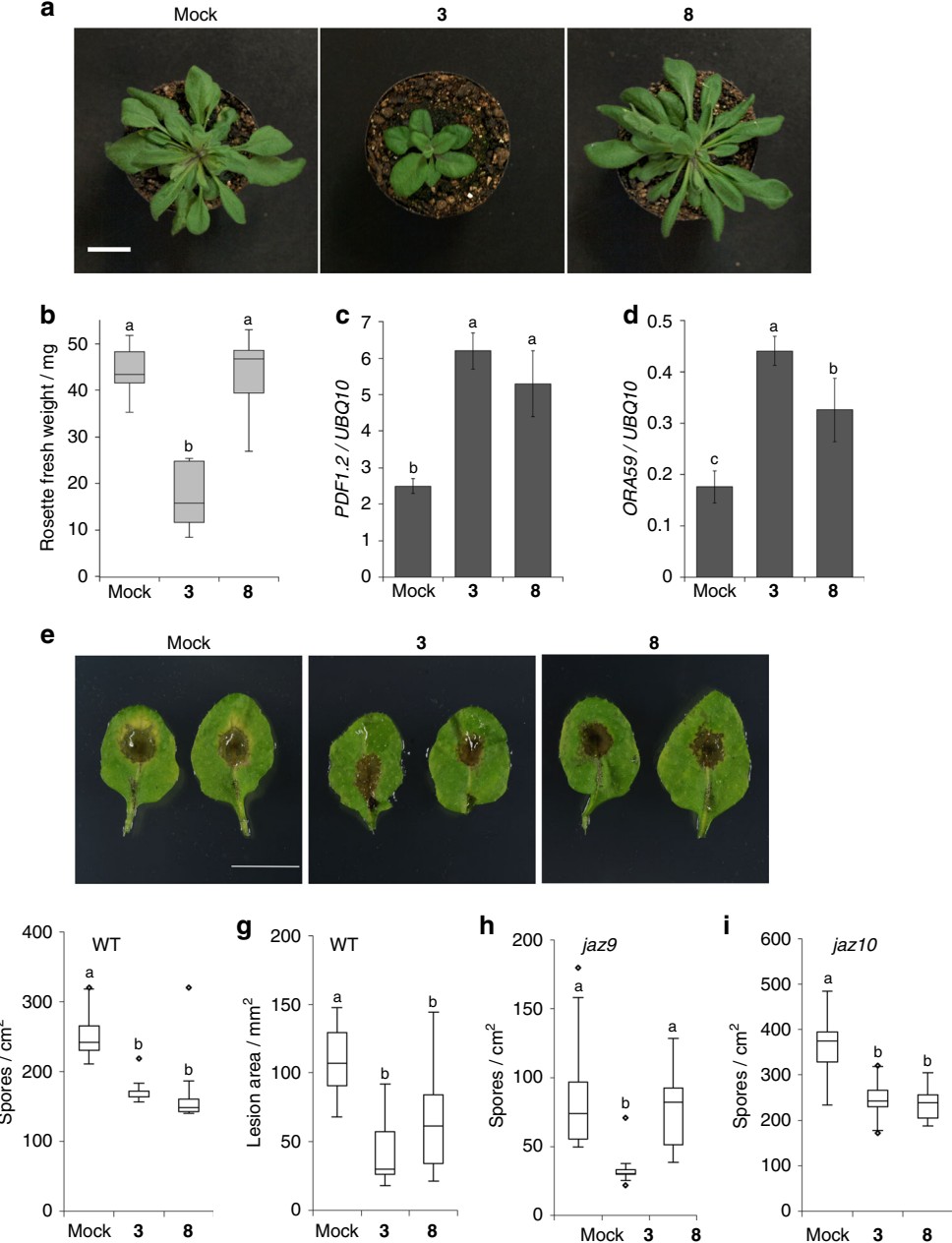

**Fig. 5** Effects of **3** and **8** on plant growth and defense of adult *Arabidopsis*. **a–d** The effects of the repetitive treatment of the compounds (**3** or **8** at 50 μM, 5 times per week from 1-week-old to 5-week-old plants) in the aerial part of WT *Arabidopsis* adult plants grown for 5-week-old plants (see Supplementary Methods). Scale bar, 2 cm. The representative images of plants treated repetitively with 50 μM of each compound (**a**), fresh weights of aerial parts of the ligand-treated adult plants (**b**), and gene expression level of the aerial parts of the ligand-treated adult plants (**c**; *PDF1.2*, **d**: *ORA59*) ($n = 7$). Significant differences were evaluated by one-way ANOVA/HSD post hoc test ($p < 0.05$). Experiments were repeated three times with similar results. **e–g** Wild-type Col-0 plants were treated with mock solution, **3** or **8** (50 μM) ($n = 11–20$) and infected with *A. brassicicola*. Representative leaves of plants infected with *A. brassicicola* are shown in **e**. Scale bar, 10 mm. Quantification of fungal spores (**f**) and lesion area (**g**) was undertaken 6 days after inoculation; the results are depicted using box-plots; horizontal lines are medians, boxes show the interquartile range and error bars show the full data range. Outliers are indicated as circles. Asterisks above columns indicate significant differences compared to the mock-treated values evaluated by one-way ANOVA/Tukey HSD post hoc test ($p < 0.05$). Experiments were repeated three times with similar results. **h–i** *jaz9* and *jaz10* mutant plants were treated with mock solution, **3** or **8** (50 μM) ($n = 14–20$) and infected with *A. brassicicola*. Quantification of fungal spores was undertaken 7 days after inoculation; the results are depicted using box-plots as described in **f**. Asterisks above columns indicate significant differences compared to the mock-treated values analyzed by one-way ANOVA/Tukey HSD post hoc test ($p < 0.05$)

The expression of *PDF1.2* and additional *ORA59*/*ERF*-regulated defense response genes plays a crucial role in plant defense against fungal pathogens[6]. Therefore, we assessed the effect of **8**, on plant defense against the fungus *Alternaria brassicicola,* as well as on growth of the adult plants. As shown in

Fig. 5a, b, repetitive addition of **3** induces strong growth inhibition on the aerial part of 5-week-old plants, whereas **8** did not. In contrast, both **3** and **8** activate the gene expression of *PDF1.2* or *ORA59* in the same way (Fig. 5c, d). Subsequently, treatment of the plant with COR (**3**) induced plant resistance

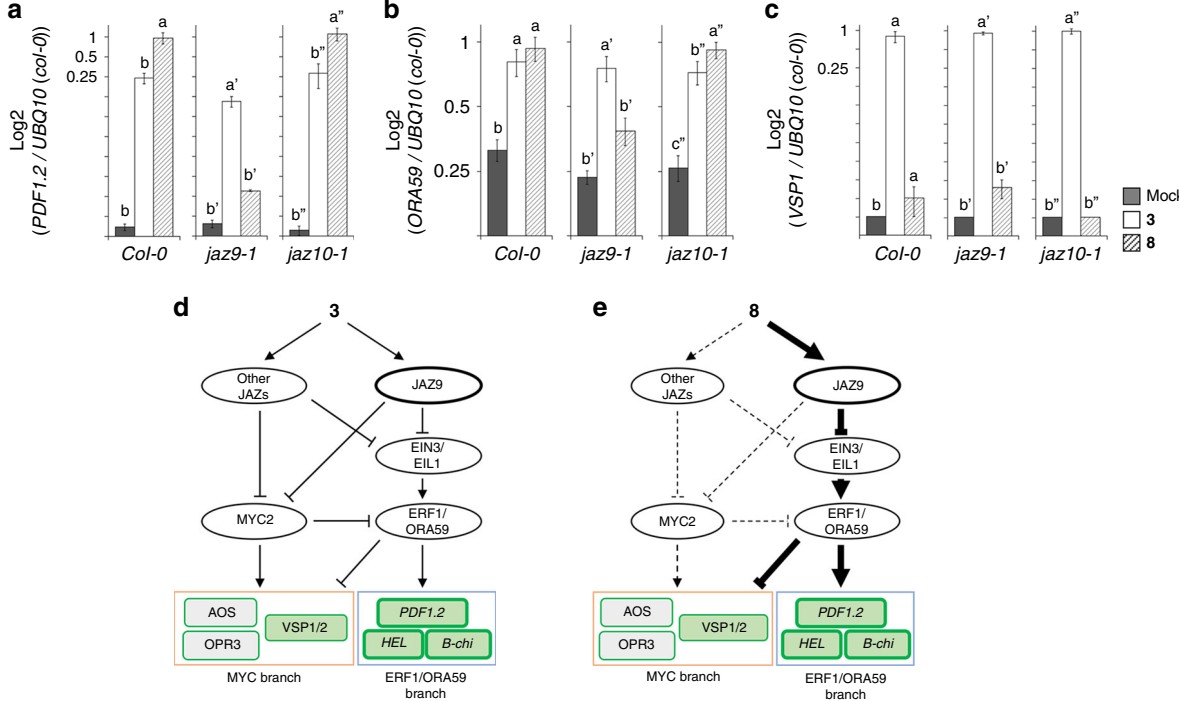

**Fig. 6** Selectively activate JAZ2-EIN3/EIL1-ORA59 signaling pathway by **8**. **a–c** Analysis of JA-responsive gene expression by qRT-PCR in 6-d-old WT (Col-0), *jaz9-1*, or *jaz10-1 Arabidopsis* seedlings with or without ligands (**3** or **8**, 1 µM) treatment (*PDF1.2* (**a**), *ORA59* (**b**), or *VSP1* (**c**)). Results shown are the mean with s.d. (n = 4). Significant differences were evaluated by one-way ANOVA/Tukey HSD post hoc test (p < 0.05). **d, e** Predicted signaling pathways involving the role of **3** (**d**) or **8** (**e**) for MYC-branch and ERF1/ORA59-branch gene expression

against the fungal pathogen compared to the mock treatment (Fig. 5e-g). Similarly, plants treated with **8** showed less chlorosis and harbored fewer fungal spores compared to the mock-treated plants after *A. brassicicola* infection which can be attributed to the upregulation of *PDF1.2* in the leaves (Fig. 5c-g). These **8**-induced resistances were impaired in *jaz9*, whereas not affected in *jaz10* (Fig. 5h-i). These results suggested that **8** can trigger comparable plant defense responses to **3**.

We then examined the MOA of **8** using *Arabidopsis* knockout mutants. Specifically, we used *jaz9* and *jaz10* knockout mutants to investigate which JAZ subtype is responsible for the **8**-mediated JA responses (Fig. 6 and Supplementary Fig. 14). JAZ9/10-selective agonist **8** upregulated the expression of *PDF1.2* in *jaz10*, whereas **8** failed to trigger this gene in *jaz9* (Fig. 6a). The knockout of *jaz9* and *jaz10* affected the **8**-induced expression of *ORA59*: **8** induces *ORA59* expression in *jaz10* as much as in WT control plants, whereas **8**-induced *ORA59* expression is completely impaired in *jaz9* (Fig. 6b). The **8**-mediated upregulation of *PDF1.2/ORA59* expression was also impaired in *jaz1/3/4/9/10* quintuple mutant (Supplementary Fig. 15a, b)[47], or in *coi1-1* mutant (Supplementary Fig. 15c, d)[48]. Additionally, **8** did not affect the root growth in *jaz9/jaz10* mutants similar to WT plants, whereas all plants still responded to **3** or *ent***6** (Supplementary Fig. 14a, b). At higher concentration, **8** triggered growth inhibition, and this was suppressed in *jaz9* mutant compared to WT or *jaz*10 (Supplementary Fig. 14c, d). Thus growth inhibition by higher concentrations of **8** may be attributed to a weak effect of the high concentrations in other COI1-JAZ co-receptors. However, these results demonstrate that **8** is mainly active on COI1/JAZ9 in planta. The **8**-triggered expression of *ERF/ORA59/ PDF1.2* was also impaired in the *ein2-1* mutant because ETHYLENE INSENSITIVE 2 (EIN2) plays an important role for the activation of ETHYLENE INSENSITIVE 3 (EIN3) and EIN3-LIKE1 (EIL1) TFs, which are responsible for the expression

of downstream genes including *ERF1*, *ORA59*, and *PDF1.2* (Supplementary Fig. 16a–c)[40,49–53]. Thus, JAZ9 is responsible for the **8**-mediated upregulation of *PDF1.2* through induction of *ORA59* regulated by EIN3/EIL1 TFs. In contrast, activation of MYC-branch genes (*VSP1* and *OPR3*, as well as *MYC2*) triggered by **8** was significantly enhanced in *ein2-1* background compared to WT plants (Supplementary Fig. 16d–f). This activation can be attributed to the release of the ERF1/ORA59 branch in this mutant. These results suggest that the JAZ9/10-selective agonist **8** upregulates the in planta expression of genes involved in JA-mediated defense responses against necrotrophic pathogen infection through the induction of PPI between COI1 and JAZ9.

## Discussion

Dissecting the growth-defense trade-off is essential if the defense responses of plants against pathogens and herbivorous insects are to be practically enhanced. Our growing knowledge of JA signaling and the identification of JA signaling components make it possible to manipulate the growth-defense trade-off in *Arabidopsis*.

In a pioneering study designed to uncouple the growth-defense trade-off in *Arabidopsis*, Campos *et al.* identified a unique *Arabidopsis* knockout mutant in which a quintet of JAZ repressors (JAZ1/3/4/9/10) and photoreceptor phyB were impaired. This mutant showed selective enhancement of defense responses against herbivores but no growth inhibition[47], representing a highly successful genetic approach for uncoupling the growth-defense trade-off. He *et al.* utilized a protein engineering approach in which the newly designed COI1[A384V] was used to uncouple the intrinsic JA signaling pathway from the **3**-mediated pathogen defense response by hijacking the JA signaling cascade to reinforce the plant defenses against pathogenic infection. COI1[A384V] was designed to have high affinity for **2** and reduced

affinity for **3** due to a point mutation in the ligand-binding pocket. Transgenic *Arabidopsis* expressing COI1[A384V] enhanced JA-mediated defense responses against insects, as well as resistance to biotrophic/hemibiotrophic pathogens that produce **3**[54]. However, both approaches require the genetic modification of plants.

In contrast, in the current study, we developed an innovative JAZ-based chemical approach for uncoupling the growth-defense trade-off of *Arabidopsis*. We rationally designed and developed the JAZ9/10 subtype-selective agonist **8**, which allowed us to use a chemical approach to uncouple the growth-defense trade-off in *Arabidopsis*. This unique agonist does not affect the growth in *Arabidopsis* seedlings (Fig. 4a, b), and selectively upregulates the expression in planta of genes involved in JA-mediated defense responses against infection by necrotrophic pathogens (Fig. 4h) by inducing PPI between COI1 and JAZ9 (Fig. 3e–g). In adult *Arabidopsis*, **8** induced enhanced resistance to the fungal pathogen *A. brassicicola*, similarly to **3** (Fig. 5a, b).

Based on our results, the MOA of **8** in *Arabidopsis* might be as follows. JA-signaling leading to defense responses relies on two branching pathways of mutually antagonistic interactions: the MYC2 branch leads to resistance against wounding and the ETHYLENE RESPONSE FACTOR (ERF) branch leads to resistance against pathogenesis (Fig. 6e)[6]. Additionally, the well-known antagonistic interaction between ORA59 (in the ERF branch) and MYC2 functions in the crosstalk between ethylene and jasmonate signaling[20,40,55–57].

Importantly, the JAZ9/10 subtype-selective agonist **8** had little effect on the expression of *VSP1*, a marker gene belonging to the MYC branch (Fig. 4j). Instead, the agonist **8** upregulates the expression of *ORA59* and *ERF1* (Fig. 4i and Supplementary Fig. 12a), and then activates the ERF1/ORA59 branch (as indicated by the increased expression of *PDF1.2*, *HEL* or *B-chi* in Fig. 4h and Supplementary Fig. 12a) and suppresses the MYC branch (as indicated by the lack of induction of *VSP1*)[43]. Guo *et al.* reported that EIN3 and EIL1, which upregulate the expression of *ORA59*, physically interact with a number of JAZ proteins including JAZ1, JAZ3 and JAZ9[55]. Therefore, our finding that **8** activated the ERF branch could be attributed to the selective activation of the JAZ9-EIN3/EIL1-ERF1/ORA59 signaling pathway. Similarly, an elicitor excreted by *Pieris rapae* caterpillars activates the ERF branch to confer resistance to necrotrophic pathogens, although the exact MOA is unknown[58]. Our potential MOA was further confirmed using *jaz9* and *jaz10* knockout mutants: the **8**-triggered upregulation of *PDF1.2* and *ORA59* was impaired in *jaz9* (Fig. 6a, b), whereas the expression of *VSP1* was slightly upregulated (Fig. 6c) due to its release from suppression by the ERF1/ORA59 branch. In contrast, the **8**-mediated expression of *PDF1.2* and *ORA59* were maintained in the *jaz10* mutant (Fig. 6a, b). It was unexpected that constitutive activation of *PDF1.2* expression was not observed in *jaz9* mutant (Fig. 6a) because the impairment of JAZ9 will release the suppression of ERF1/ORA59 branch[55]. Moreover, **8**-mediated transcriptomic changes corresponding to metabolism and regulation observed in WT and *jaz10* were dramatically suppressed in *jaz9* mutant (Supplementary Fig. 17). One possible explanation is that the unidentified repression of EIN3/EIL1 by other JAZ subtypes may occur in the *jaz9* mutant. For instance, whereas JAZ9 could be a major repressor of EIN3/EIL1 compared to other JAZ, in the absence of JAZ9 (in the *jaz9* mutant) other JAZ could occupy EIN3/EIL1 compensating the JAZ9 repression. This hypothesis is supported by the following results: the treatment by **3** which cause degradation of all JAZ subtypes upregulated the expression of *PDF1.2* (Fig. 6a) and *ORA59* (Fig. 6b) in *jaz9*, which was not observed by treating with **8**. Campos *et al.* also reported that the *PDF1.2* expression was not upregulated in the *jaz1/3/4/9/10*

quintuple mutant[47], suggesting that functionally redundant remaining JAZ subtypes may be involved in the repression of EIN3/EIL1 in the absence of JAZ9[59].

Overall, combined with the specific JAZ degradation using JAZ-GUS reporter lines (Fig. 3f–g), **8** mainly induces the degradation of JAZ9 through the activity of F-box protein COI1, which results in the selective activation of EIN3/EIL1, and subsequently ERF1/ORA59, the deactivation of MYC2 and ultimately, the upregulation of *PDF1.2*, which is involved in defense responses against necrotrophs. Thus, this defense response is selectively enhanced and do not cause growth inhibition in planta (Fig. 6d, e). The development of a chemical regulator able to promote plant defense but having no growth inhibition has important agricultural applications especially for crops unamenable to genetic modification. Recently, JA-macrolactones, synthetic analogs of **2**, were also reported to uncouple growth and defense responses against herbivores in wild-type *Nicotiana attenuata*[60]. Although the MOA of the molecules remains unclear, chemically modulating the ligand activity of Jas is a promising approach for modulating various activities of this phytohormone to uncouple plant growth and defense responses in non-model plants in the future.

In summary, we have rationally developed an agonist **8** stabilizing JAZ9/10-selective PPI for the COI1-JAZ co-receptor via the in silico docking studies using a stereochemical isomer of **3**. The agonist **8** uncoupled the growth-defense trade-off in *Arabidopsis* to upregulate the expression of *PDF1.2*, a key defense gene involved in responses against necrotrophic pathogens. This selective activation can be attributed to the selective activation of the ERF branch of the JA signaling pathway through JAZ9-EIN3/EIL1- ERF1/ORA59. To the best of our knowledge, this is the first example of a JA-Ile agonist with high selectivity for specific protein subtypes. Recent study implies that different JAZ members might be responsible for a specific function: JAZ2 is specifically expressed in guard cells to control the stomatal response during bacterial invasion[61]. The JAZ9/10-selective PPI agonist **8** is expected to serve as an important chemical tool for regulating the plant growth-defense trade-off and for elucidating the complex regulation of JA signaling pathway in plants. It may serve as a lead molecule for the development of commercial products that are able to enhance plant disease resistance with limited penalty on growth.

## Methods

**General materials and methods**. All chemical reagents and solvents were purchased from commercial suppliers (Kanto Chemical Co. Ltd., Wako Pure Chemical Industries Co. Ltd., Nacalai Tesque Co., Ltd., Enamine Ltd., Watanabe Chemical Industries Co. Ltd., or GE Healthcare) and used without further purification. Reverse-phase high-performance liquid chromatography (HPLC) was conducted on a PU-4180 plus with UV-4075 and MD-4010 detectors (JASCO, Tokyo, Japan). UV detection was performed at 220 nm. [1]H and [13]C NMR spectra were obtained on a JNM-ECS-400 or JNM-ECA700 spectrometer (JEOL, Tokyo, Japan) in CD$_3$OD. Fourier transform infrared (FT/IR) spectra were recorded on a FT/IR-4100 (JASCO, Tokyo, Japan). High-resolution (HR) electrospray ionization (ESI)-mass spectrometry (MS) analyses were conducted on a microTOF II (Bruker Daltonics Inc., Billerica, MA). MALDI-TOF MS analyses were carried out on a 4800 plus MALDI TOF/TOF Analyzer (AB Sciex, Framingham, MA). SDS-PAGE and western blotting were analyzed with a Mini-Protean III electrophoresis apparatus (Bio-Rad, Hercules, CA). Chemiluminescent signals were detected with an LAS 4000 imaging system (Fujifilm, Tokyo, Japan). The 3D structures of the ternary complex shown in Fig. 3a-c and Supplementary Fig. 5 were constructed using MOE 2016.08 software (Chemical Computing Groups, Montreal, Canada).

**Pull-down experiments using full-length JAZ proteins**. The plasmids of GST-fused COI1 or ASK1 (pFB-GTE-COI1 and pFB-HTB-ASK1) were obtained from Addgene (https://www.addgene.org/). These proteins were co-expressed in insect cells and purified by Glutathione Sepharose 4B (GE Healthcare)[17,30]. MBP-fused JAZ proteins (JAZ1 and JAZ3)[10,26] were expressed in *Escherichia coli* cells (BL21 (DE3)) and purified by amylose resin (New England Biolabs)[10,26,30]. In each pull-down experiment, purified COI1-GST (5 nM) with ASK1 and coronatine analogs

(100 nM) are dissolved in 500 μL of buffer for pull down experiments (50 mM Tris-HCl, pH 7.8, containing 100 mM NaCl, 10% glycerol, 0.1% Tween20, 20 mM 2-mercaptoethanol, 100 nM IP$_5$, EDTA-free complete protease inhibitor cocktail (Roche)) and added to amylose resin-bound MBP-JAZ (25 μL suspension of amylose resin containing 40 nM of MBP-JAZs (JAZ1 and JAZ3). After 4 h incubation at 4 °C under rotation, the amylose resin were washed in triplicate with 500 μL of fresh buffer, and then was resuspended with SDS-PAGE loading buffer containing maltose (20 mM, 50 μL). After boiling for 10 min at 60 °C, the samples were loaded on SDS-PAGE and analyzed with western blotting. The bound COI1-GST were detected using anti-GST HRP conjugate (RPN1236, GE Helthcare, 2500-fold dilution in skimmed milk solution). MBP-JAZ were detected using rat anti-MBP antibody (016–24141, Wako, 5,000-fold dilution in phosphate buffered saline (PBS) containing 0.1% tween 20) and goat anti-rat IgG-HRP antibody (sc-2032, santa cruz biotechnology, 40,000-fold dilution in PBS containing 0.1% tween 20). Uncropped blots of Fig. 1d,e were shown as Supplementary Fig. 18.

**Pulldown experiments using epitope-conjugated JAZ peptides.** For the pull-down experiment, purified COI1-GST (5 nM), OG-conjugated JAZ peptide (10 nM), and coronatine analogs (100 nM) of incubation buffer (50 mM Tris-HCl, pH 7.8, containing 100 mM NaCl, 10% glycerol, 0.1% Tween20, 20 mM 2-mercaptoethanol, and 100 nM IP$_5$)[10,17,30,62] were combined with anti-fluorescein antibody (Abcam, ab19491, 0.25 μL) and incubated for 10–15 h at 4 °C with rotation. After incubation, the samples were combined with Protein A Mag Sepharose Xtra (GE Healthcare, 25 μL in 50% incubation buffer slurry). After 3 h incubation at 4 °C with rotation, the samples were washed three times with 500 μL of fresh incubation buffer. The washed beads were resuspended in 50 μL of SDS-PAGE loading buffer containing dithiothreitol (DTT, 100 mM). After heating for 10 min at 60 °C, the samples were subjected to SDS-PAGE and analyzed by western blotting. The bound COI1-GST proteins were detected using anti-GST HRP conjugate (RPN1236, GE Healthcare, 2500-fold dilution in skimmed milk solution). Uncropped blot of Fig. 2c were shown as Supplementary Fig. 19. Uncropped blot of Fig. 2d,e and Supplementary Fig. 3 were shown as Supplementary Fig. 20. Uncropped blot of Fig. 3e were shown as Supplementary Fig. 21. Uncropped blot of Supplementary Fig. 4a were shown as Supplementary Fig. 22. Uncropped blot of Supplementary Fig. 6 were shown as Supplementary Fig. 23.

**In silico docking study.** The initial structure of the COI1-3-JAZ1 complex was obtained based on its crystal structure (PDB ID: 3OGM). MODELLER was used to deduce the structures of the absent residues (residues 68–79 and 550–562) of COI1. The model structures of JAZ3, 9, 10, 11, and 12 were constructed by mutating residues of the JAZ1 peptide of the COI1-3-JAZ1 complex with Swiss-PdbViewer. We then prepared the structures of COI1-ent6-JAZ3 and COI1-ent6-JAZ12 by replacing 3 with ent6 by docking simulation. Also, the structures of COI1-8-JAZ9, and COI1-8-JAZ10 were obtained by replacing ent6 of the equilibrated COI1-ent6-JAZ9 and COI1-ent6-JAZ10 structures, obtained from the molecular dynamics (MD) simulations. Docking simulations of compounds were performed using DOCK 6.6 software. Amber99 force field parameters were assigned for the estimations of grid score. The anchor-and-grow algorithm was used to search for the best docked ligand conformations. The space of the conformation search was defined in a 16 Å radius from the center of binding pocket of COI1. The grid volume was determined to cover the ligand search space with spacing between the grid points maintained at 0.3 Å. The obtained docked poses, which showed the best score for each model, were adopted for subsequent MD simulations of the complex models in water solvent. Five independent 100 ns-long MD simulations with different initial velocities for the COI1-3-JAZ (JAZ1, JAZ3, JAZ9, JAZ10, JAZ11, and JAZ12) models were carried out to sample the binding structures of 3 in each model. Similarly, five 100 ns-long MD simulations of the COI1-ent6-JAZ models (JAZ3, JAZ9, JAZ10, JAZ11, and JAZ12) and COI1-8-JAZ models (JAZ9 and JAZ10) were performed for total of 500 ns. It should be emphasized that no water molecules were observed in the binding pocket of either the COI1-3-JAZ1 crystal structure, or the snapshots obtained from the MD simulations. Therefore, the role of water molecules is not discussed in this study. All MD simulations were run under conditions of constant temperature and pressure ($T$ = 300 K, $P$ = 1 atm). A Parrinello-Rahman type thermostat[63] and a Nosé-Hoover type barostat[64] were used to control system temperature and pressure. The force field parameters of Amber03[65], generalized amber force field (gaff)[66], and TIP3P water model[67] were assigned for the protein, ligand, and water molecule, respectively. The cutoff for the van der Waals (vdW) interaction was 12 Å. The Particle mesh Ewald (PME) method[68] was used to calculate the Coulomb electrostatic interactions. The time step for integration was 2 fs. The sampled complex structures were stored every 10 ps. All MD calculations were performed using the GROMACS 5.1.4 program package. The sampled COI1-ent6-JAZ3 and COI1-ent6-JAZ12 complex structures were stored every 10 ps. Radial distribution functions (RDFs) were calculated to investigate the hydrogen bond networks between the compounds and surrounding residues in the binding pocket of COI1. The RDF curves for possible hydrogen bond pairs in each model were evaluated using total 500 ns term MD simulation. The first peak position of RDF curve can be used for judgment of formation of hydrogen bond during the MD simulations; in this study, we identified the formation of hydrogen bond when the first peak position of RDF was in 2.5 Å[69]. All RDF calculations were done by VMD 1.9.3.

**Plant material and growth conditions.** A. thaliana ecotype Col-0 seeds were surface-sterilized in 5% sodium hypochlorite with 0.3% Tween20 and vernalized for 2–4 days at 4 °C. All seedlings were grown under a 16 h light (118 μmol m$^{-2}$ s$^{-1}$; cool-white fluorescent light)/8 h dark cycle at 22 °C in a Biotron NC-220 growth chamber (Nippon Medical & Chemical Instruments Co., Ltd., Osaka, Japan). WT and mutant seedlings were grown in 1/2 Murashige and Skoog (MS) liquid medium. $P_{35S}$-JAZ1:GUS, $P_{35S}$-JAZ9:GUS, $P_{35S}$-JAZ10:GUS, $P_{35S}$-JAZ11:GUS, and $P_{35S}$-JAZ12:GUS seedlings were vertically grown on 1/2 MS plates for 4 days.

**Root length measurements.** Two-d-old seedlings of Col-0 were transferred on 1/2 MS plate in the presence or absence of 0.1–10 μM of each compound, and grown under 16 h light/8 h dark cycle at 22 °C in growth chamber for 4 days. Then, root length of each seedling was measured. Images were taken with an E-520 digital camera (Olympus Corp., Japan), and root length was measured using Image J 1.45S software (http://imagej.net/Welcome).

**Measurement of fresh weights and accumulated anthocyanin.** Seedlings were germinated on 1/2 MS plate for 2 days and were transferred on 1/2 MS plate in the presence or absence of the 0.1–10 μM of each compound and grown under a 16 h light/8 h dark cycle at 22 °C in growth chamber for 4 days. 12–18 seedlings were cut at the base of the leaf and weighed. For each sample, seedlings were homogenized with aqueous methanol (HCl 0.1%; 50% methanol/sterilized water (v/v)), and then incubated at 4 °C for 3 h. The samples were soaked in chloroform for the extraction of anthocyanins. Total anthocyanins were determined by measuring the $A_{530}$ and $A_{657}$ of the aqueous phase using a spectrophotometer (NanoPhotometerN60, IMPLEN) and the content was calculated from $A_{530} - 0.25 \times A_{657}$ per fresh weight (mg)[70,71]. Values represent mean ± s.d.

**Quantitative RT-PCR analyses.** In a Biotron NC-220 growth chamber (Nippon Medical & Chemical Instruments Co., Ltd., Japan), each filtered compound was treated to six day old plants in autoclaved 1/2 MS liquid medium under a 16 h light (118 μmol m$^{-2}$ s$^{-1}$; cool-white fluorescent light)/8 h dark cycle at 21 °C. Ligands (3, ent6 or 8, 0.1–10 μM) were treated for 2 h (early-term JA-responsive genes: JAZ1, AOS, OPR3, LOX3, TAT3, other genes: ORA59, MYC2, ERF1). These ligands were treated to plants for 8 h (late-term JA-responsive genes: PDF1.2, VPS1, HEL, B-chi, LOX2). Then, using an RNeasy Mini Kit (Qiagen Co. Ltd., Germany), total RNA was isolated and then first-strand cDNA was gained with ReverTra Ace reverse transcriptase (Toyobo, Japan) with oligo-dT primers. A StepOnePlus Real-Time PCR System (Life Technologies, USA) was used for quantitative PCR (all primers sequences for qPCR in Supplementary Table 1). Polyubiquitin 10 was used as a reference gene.

**GUS staining and quantification.** Four-d-old seedlings of $P_{35S}$-JAZ1:GUS, $P_{35S}$-JAZ9:GUS or $P_{35S}$-JAZ10:GUS[26] were transferred in liquid 1/2 MS containing 1 μM COR, ent6 or 8 for 30 min. Seedlings were then immersed in GUS staining buffer (50 mM phosphate buffer, pH 7.0; 3 mM K$_4$Fe(CN)$_6$ (≥99.5%, Wako Pure Chemicals, Japan); 0.5 mM K$_3$Fe(CN)$_6$ (≥99%, Wako Pure Chemicals, Japan); 20% MeOH; 1 mg ml$^{-1}$ 5-bromo4-chloro-3-indolyl β-D-glucuronic acid (X-Gluc, Bio medical science, Japan)) at 37 °C. After staining, the solution was exchanged to 70% ethanol. Images were taken under a microscope (Stemi 2000-C, ZEISS, Germany) equipped with docking digital camera (AxioCam ERc 5 s, ZEISS, Germany). Alternatively, 20 seedlings of $P_{35S}$-JAZ1:GUS, $P_{35S}$-JAZ9:GUS, $P_{35S}$-JAZ10:GUS, $P_{35S}$-JAZ11:GUS or $P_{35S}$-JAZ12:GUS were treated in liquid 1/2 MS medium with 1 μM 3, ent6 or 8 for 30 min (for JAZ1,9,10,12) or 2 h (for JAZ11). Then 20 roots were collected, frozen and were homogenized with extraction buffer (50 mM phosphate buffer, pH 7, 10 mM 2-mercaptoethanol; 0.1% sarcosyl (N-lauroylsarcosine sodium salt; > 94%, Sigma-Aldrich) and 0.1% Triton X-100). For 1 h at 37 °C, 30 μL extract were incubated with 70 μL protein extraction buffer (1 mM MUG (methylumbelliferyl-β-D-glucuronide hydrate; ≥ 98%, Sigma-Aldrich)). 10 μL samples were collected at t = 0 and t = 10 min or 1 h. Then the reaction was stopped with 90 μL 0.2 M Na$_2$CO$_3$. Using the spectrophotometer Infinite M200Pro (TECAN, Switzerland), fluorescence was detected at ex/em 365/460 nm (n = 4, values represent mean ± s.d.). Three independent replicates were measured with similar results.

**Quantification of endogenous glucosinolate.** Four seedlings (15–25 mgFW) were homogenized. Then, 4-methylthiobutyl glucosinolate (4MTB) were extracted from the sample using 1 mL of 28% (v/v) aqueous acetonitrile with 0.05% acetic acid. The mixture was incubated overnight in dark at 4 °C and then centrifuged at 20,000×g for 5 min. 900 μL of the supernatant was collected. At room temperature the liquid was dried with nitrogen gas flow and then added in 40 μL of ultrapure water. 10 μL of sample was subjected to Ultra-performance liquid chromatography coupled with time-of-flight mass spectrometry (UPLC-TOFMS) analysis on an Agilent 1290 Infinity (Agilent Technologies, USA) coupled with a micrOTOF II (Bruker Daltonics, Germany). We used a ZORBAX Eclipse Plus C18 column (1.8 μm, 2.1 × 50 mm; Agilent Technologies) for the analyses of compounds on UPLC (the mobile phases: A, 20% (v/v) aqueous acetonitrile with 0.05% (v/v) acetic acid; B, acetonitrile with 0.05% (v/v) acetic acid: the gradient program: 0 to 3.5 min, isocratic 90% A; 3.5 to 6 min, linear gradient 90 to 0% A; 6.1 min to 9 min, isocratic

90% A, with a flow rate of 0.15 mL min$^{-1}$). The mass spectrometry analyses were carried out on a negative mode (scan range of 100–700 $m/z$) under the following conditions: the capillary voltage = 4,200 V, the nebulizer gas pressure = 1.6 bar, the desolvation gas flow = 8.0 L min$^{-1}$, the temperature = 180 °C.

**Microarray analysis**. Six-day-old plants for microarray analysis were incubated in autoclaved 1/2 MS liquid medium containing each filtered compound under a 16 h light (118 µmol m$^{-2}$ s$^{-1}$; cool-white fluorescent light)/8 h dark cycle at 21 °C in a Biotron NC-220 growth chamber (Nippon Medical & Chemical Instruments Co., Ltd., Japan). The four plants were treated with or without ligands (**3**, *ent***6** or **8**, 1 µM) for 8 h in the same growth chamber (Three replicates for each treatment). After treatments, total RNA was isolated with an RNeasy Mini Kit (Qiagen Co. Ltd., Germany). Microarray analyses were carried out as described previously with few modifications[72]. Briefly, total RNA (400 ng) was labeled with fluorescently labeled Cy3, using a Quick Amp labeling kit (Agilent Technologies) and resulting cRNA was subsequently hybridized to an Agilent *Arabidopsis* custom microarray (GPL19830). Microarray analyses were performed with three biological replications. Arrays were scanned with a microarray scanner (G2505B, Agilent Technologies) and microarray data were processed and analyzed using GeneSpring GX (v.14.9, Agilent Technologies) with quantile normalization. Statistical significance was assessed using a one way ANOVA with BH correction[73] and a 95% confidence interval (Corrected *P*-value < 0.05). A Tukey's HSD (honest significant difference) test with BH correction was performed as a post hoc test (Corrected *P*-value < 0.05). Genes with false discovery rate (FDR) values less than 0.05 and at least 2.5-fold changes were regarded as up- or down-regulated. Heat map analyses were performed with an online tool heatmapper[74]. The normalized log$_2$ values were then used to compare the transcriptomic changes using MapMan 3.6.0RC1[75]. Gene ontology enrichment analyses were carried out using the PANTHER (protein annotation through evolutionary relationship) classification system database maintained at http://pantherdb.org/[76].

**Repetitive chemical treatment of *Arabidopsis* seedlings**. The leaf of six-day-old wild-type Col-0 plants were treated with 2 µL of aqueous solution containing the 0–10 µM of each compounds. After incubation for three days, each compound was applied again in a same way. After 8 h incubation, images were taken with an E-520 digital camera (Olympus Corp., Japan), then the root was cut off and the residual aerial part was weighed (total 9-day-old plants) ($n = 5$, values represent mean ± s. d). Three independent replicates (seedling) were measured with similar results.

**Repetitive chemical treatment of adult plants of *Arabidopsis***. Wild-type Col-0 seedlings were germinated on 1/2 MS plate for six days and the leaf were treated with 2 µL of aqueous solution containing the mock solution or 50 µM of **3** or **8**. After incubation for three days (total 9 days), each compound was applied again in a same way. After incubation for four days (total two weeks), seedlings were transferred in soil and the plants were again treated with 2 µL drop per leaf and in the rosette center with mock solution, 50 µM of **3** or **8**. After incubation for 1 week (total three weeks), and for two weeks (total 4 weeks), the plants were again treated with 6 µL drop per leaf and in the rosette center with mock solution, 50 µM of **3** or **8**. Finally, the 5-week-old plants were again treated with 6 µL drop per leaf and in the rosette center with mock solution, 50 µM of **3** or **8**, then after 8 h incubation, images were taken with an E-520 digital camera (Olympus Corp., Japan), then the root was cut off and the residual aerial part was weighed (total 5-week-old plants) ($n = 7$, values represent mean ± s.d), and then the plants were immediately frozen in liquid nitrogen for qRT-PCR experiments. Three independent replicates (seedling) were measured with similar results. The frozen plants were pound and lyzed in a mortar, and isolation of total RNA, preparation of first-strand cDNA and qRT-PCR were performed in the same way as shown in the Online Methods ("quantitative RT-PCR analyses").

**Fungal infection analyses**. We used soil-grown plants for fungal infection. Wild-type Col-0 plants were treated with 6 uL fungal spore suspension in the rosette center of each leaf with mock solution, 50 µM **3** or **8** 7 h before infection, concurrently and three days after fungal infection. At least 15 leaves of four-week-old plants (3 leaves/plant) were inoculated with 20 µL of a suspension of 10$^6$ *A. brassicicola* spores/ml PDB (Difco)[77]. Disease symptoms were quantified by photo-images taken 6 to 8 days after inoculation and lesion diameter was quantified in twelve to fourteen leaves of six different plants for each treatment using the ImageJ software. Spores were quantified in a hemocytometer under a light microscope (Leica DMR UV/VIS). Five inoculated leaves of five different plants were pooled for each biological sample, and four to eight independent biological replicates were measured for each treatment. All data were analyzed by one-way ANOVA/Tukey HSD post hoc test ($p < 0.05$). This experiment was repeated three times with similar results.

**Data availability**
The microarray data has been deposited to GenBank with the accession number GSE110858. The authors declare that all other data supporting the findings of this study

are available within the article and its Supplementary Information files or are available from the corresponding author upon request.

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

## Acknowledgements

We thank Prof. E. Kombrink (Max Planck Institute for Plant Breeding Research, Köln) for providing us with seeds for the *jar1* mutant, and Mr. K. Hayashi, Mr. T. Iwashita, and Mr. K. Julien (Tohoku Univ.) for technical assistance. The seed of *jazQ* was kindly provided by Prof. Gregg Howe (Michigan State Univ.). This work was supported by a Grant-in-Aid for Scientific Research on Innovative Areas (no. 23102012) 'Chemical Biology of Natural Products (no. 2301)' for M.U. from MEXT, Japan, a Grant-in-Aid for Scientific Research (no. 26282207, 17H06407, and no. 17H00885 to M.U.), JSPS A3 Foresight Program (to M.U.), Naito Foundation (to M.U.), JST (JPMJPR16Q4 to Y.T.), JST CREST (JPMJCR13B4 to M.S.), and Spanish Ministry for Science and Innovation grant BIO2016-77216-R (MINECO/FEDER) for A.C. and R.S. This study was supported by JSPS KAKENHI (JP16K05648 to H.S.). The computations in this study were performed using the advanced center for computing and communication of RIKEN, the research center for computational science of Institute for Molecular Science (IMS), the research center for advanced computing infrastructure of Japan Advanced Institute of Science and Technology (JAIST), and the center for computational science of Tsukuba university.

## Author contributions

M.U. conceived and designed the research project and interpreted all results. M.U. Y.T., Y.I., A.C. and R.S. wrote the manuscript. Y.T. designed the in vitro PPI assay. Y.T. and M.I. performed protein assays, qRT-PCR analyses, and phenotype assays. A.C. and R.S. interpreted the results and performed in planta experiments. H.S. performed in silico docking studies. S.E. and N.K. synthesized chemicals. Y.I. designed and interpreted the results of gene expression analyses. M.T., K.B., and M.S. carried out DNA microarray analyses.

## Additional information

**Competing interests:** The authors declare no competing interests.

