## [Peer Review File · Nature Communications]

Reviewers' comments:

Reviewer #1 (Remarks to the Author):

In this study, the authors developed an agonist of the plant hormone jasmonate (JA) with the objective of eliciting JA-dependent defense responses without compromising plant growth. Based on the structure of the JA-Ile mimic coronatine (COR), they first synthesised 6 compounds and tested them for their capacity of forming COI1-JAZ receptor complexes. One compound, ent6, was able to selectively induce complex formation between degron domains of JAZ3/9/10/11/12 and COI1 but not with JAZ1/4/5/6 in a *E. coli* pull-down system. This prompted the authors to develop 3 additional molecules based on the structure of ent6 that could selectively form receptor complexes between COI1 and specific JAZ repressors only. With the same *in vitro* pull-down technique, the authors identified compound 8 (O-phenyl oxime 8), that could specifically trigger receptor formation between COI1 and JAZ9/10, but not with other JAZs. To test the function of compound 8 in planta, the authors used 35Sp:JAZ-GUS transgenic lines to show that treatment with 8 triggers JAZ9 and JAZ10 degradation but not the degradation of JAZ3. Importantly, as opposed to COR treatments that stunt root growth, treatments with compound 8 did not have a major effect on root length. However, treatments or pre-treatments with compound 8 induced defense gene expression and rendered the plants more resistant to the fungal pathogen *Alternaria brassicola*. Overall, the authors claim that compound 8 is an elicitor of selective COI1-JAZ9 and COI1-JAZ10 interactions (and subsequent JAZ degradation) that results in defense elicitation but no growth repression. This lack of the classical growth/defense trade-off is supposedly triggered by the downstream de-repression of JAZ9-repressed transcription factors EIN3/EIL1-ORA59.

The development of compound 8 is interesting for potential agronomic applications and possible analysis of specific signalling networks governed by individual JAZs (in this case JAZ9 and JA10). Previous studies have also addressed the question of how to disengage the JA-mediated growth-defense trade-off (Campos et al 2016 *Nat Commun*) and engineered alternative COI1-JAZ ligands for biotech applications (Monte et al 2014 *Nat Chem Biol*), as well as ligands for the auxin receptor TIR1 that is structurally very similar to COI1. At the mechanistic level, this work is mostly an extension of the work done by Monte et al 2014 *Nat Chem Biol*. The manuscript is overall written and presented in a concise, clear and logical way.

However, several claims in this work are very weak and/or remain unsubstantiated:

1. A major claim in the paper is that treatments with compound 8 do not affect plant growth. This is supported by a single root length experiment with <20 individuals per genotype and a large variability (Fig4a,b). The authors should perform a proper dose-response growth analysis both in roots and aerial organs (eg. repetitive spraying), and compare the effects of compound 8 with effects elicited by COR. Importantly, those effects should also be tested in *jaz9/jaz10* mutants that should theoretically (if the authors' claims are correct and compound 8 is active only on COI1-JAZ9/10) result insensitive to compound 8 and still respond to COR or JA-Ile. Similarly, what are the long-term physiological consequences of compound 8 treatments?

2. No evidence is presented regarding potential off-target effects of compound 8. How specific is the effect of this new ligand? Does this small molecule induce any other changes in addition to COI1-JAZ9/10 interactions? First, the specificity of 8 vs ent6 on JAZ degradation (Fig.3f) should be extended to other JAZs (eg. JAZ3, JAZ11, and JAZ12) in a quantitative manner. Then, the authors should analyse the global differences in transcriptome changes occurring after compound 8 and COR treatments and evaluate the effects occurring in WT vs *jaz9/10* mutants. Furthermore, is compound 8 able to specifically induce COI1-JAZ binding but not TIR1-AUX/IAA binding?

3. In addition to point 1 and 2, this work would be particularly strengthened if the authors could show that compound 8 is able to induce defense responses without compromising growth in other agronomically relevant plant species (eg. rice, wheat, maize, ..)

MINOR POINTS

- Pg.4: Fig1b is cited before Fig1a and describes COR-dependent COI1-JAZ interaction. However, there is no explanation of COR in the text at this point
- Line.53: The term 'player proteins' is unclear and unspecific, just as Line.61 'dream molecule'
- Legend Fig2c: Specify what is 'Std'
- Line.163: why unexpectedly?

Reviewer #2 (Remarks to the Author):

This submission describes the synthetic design and functional analysis of a jasmonate agonist that may potentially uncouple the growth-defense trade-off that is normally encountered with the 'natural' jasmonate hormone JA-Ile. The authors attempted to create such a molecule by looking for coronatine derivatives with altered specificities to trigger JAZ-COI1 interaction. Whereas the authors have clearly developed a small molecule that has a lower affinity to 11 of the 13 JAZ proteins, the overall claims of the manuscript are not supported sufficiently by the data. Most importantly, the authors constantly use different concentrations of their molecules in the different assays, hence they do not show at all that the growth-defense trade-off is actually uncoupled. Secondly, one may wonder if the claimed JAZ-subtype-selectivity of their molecule would not rather reflect it being 'only' a weaker jasmonate molecule. Below my major comments are listed.

- The authors carry out all in planta assays at different concentrations, hence their data do not support the uncoupling of the growth-defense trade off yet. For compound 8 these are respectively 1 μM for the root growth inhibition (Fig 4a) and GUS degradation assays (Fig 3f) but 10 μM for the qRT-PCR analysis (e.g. Fig. 4d for the PDF1.2 expression induction) and even 50 μM for the fungal resistance assay. I wonder whether at a concentration higher than 1 μM of compound 8 no growth penalty would be observed, particularly given the fact that many 'early' JA-responsive genes, such as the JAZ, MYC2 and JA biosynthesis genes are clearly transcriptionally induced at 10 μM . To justify their major claim, the authors should therefore carry out all in planta assays at the 3 concentrations already used.
- Along those lines, it will also be needed to test e.g. anthocyanin and/or glucosinolate accumulation in seedlings treated with compounds 3, ent6 and 8, and see to what extent the activation of specialized metabolism is also triggered by e.g. compound 8, which would further strengthen its value and utility. Likewise, additional 'resistance' assays such as wounding and/or bioassays with caterpillars after treatments with compounds 3, ent6 and 8, should be carried out to assess the defense 'range' of the different compounds.
- Finally, the extent of an eventual uncoupling of growth and defense could be further strengthened by a genome-wide transcriptome analysis (by RNA-Seq). GO-enrichment term analysis could be carried out on such a data set. The experiment should include at least Col0 (WT) seedlings mock-treated and elicited with compounds 3, ent6, and 8.
- The designed coronatine derivatives should be tested at a higher concentration range in the in vitro co-immunoprecipitation assays, in particular compounds ent6 and 8. For instance the experiment shown in Figure S4 should actually be carried out for all of the JAZ peptides and particularly for all those seemingly not being bound by ent6, namely JAZ1-4-5-6. Likewise, this experiment should also be conducted for compound 8 for all JAZ peptides. Only then, unambiguous conclusions can be drawn on whether there is true JAZ subtype-specificity or not. Along those lines, authors may have to edit some of their statements such as on page 8 lines 130-131 where the statement of strong and weak

PPI induction by ent6 for JAZ11-12 and JAZ9-10, respectively, does not seem to be supported by the data. I do not really see a difference in Fig. 2c. Likewise, to allow assessing the specificity of compounds 7-9 correctly, compound ent6 should be included at 500nM in the assay shown in Fig 3e because now the authors compare only with the 100nM assay shown in fig 2.

- If all of the above analysis is carried out, it would be informative to map the selectivity of compounds ent6 and 8 on a phylogenetic tree of the JAZ family, which will allow judging whether there is any phylogenetic correlation in their activity/specificity.
- The JAZ-GUS degradation experiment is very nice. To support the specificity of the PPI induction further, the authors could carry out a Y2H analysis for ligand dependent PPI between COI1 and different JAZs with the LexA system.
- qRT-PCR analysis. It is not indicated which statistical analysis is conducted. Furthermore, in several cases the authors do not apply the statistical analysis correctly to allow supporting their conclusions. Most importantly, they should not only assess statistical significant differences between treatment with mock and 8 but also between treatment with 3 and 8 and with ent6 and 8 (Figures 4 and S6-S7-S8). Only then they can draw conclusions about the specificity of 8. Likewise they should also statistically compare the effects of 3 and 8 between Col-0 and the jaz mutants, and not only the effect of the compounds within the genotype (Fig. 6a-c).
- The statement that ORA59 expression is more activated by 8 than by 3 or ent6 is not true. This is actually also very clear from Fig6b! Likewise, the authors seem to cherry pick data for drawing conclusions. Whereas I agree that there is a differential trend for PDF1 expression, I fail to see a different trend between e.g. JAZ1-AOS-MYC2 on the one hand and ERF1-HEL-CHI on the other hand, contrary to the authors' view. These kinds of erroneous statements can be avoided if statistical analysis is properly conducted as suggested above. The overall trend of the expression data actually points to compound 8 being a 'weak' JA-like molecule rather than a specific one.
- The expression data with the ein2-1 mutant are also intriguing (Fig. S8). Also the effect of 3 on PDF1 and ORA59 expression is clearly down or abolished. I did not check whether this corresponds with data previously published on this mutant (if ever checked), but it would certainly be interesting to assess also the effects of 3 on the expression of other JA-response genes in this background. These data will be needed to further support the model proposed in Fig 6d-e. Eventually, this may possibly also help the authors to understand better the intriguing observation that in the jaz9 mutant, nor in the jaz10 or the quintuple jaz mutant, the expression of PDF1 is not released in the absence of JAs.

Minor comments

- What does 'Std' in Fig 2c stand for?
- Figure 5, include the quantification of the lesion diameter after *Alternaria brassicicola* infection.
- Fig. S1. It would be helpful to the reader to indicate which JAZ peptides were eventually produced for the binding assays.
- Use of the English language should be verified. There are numerous grammar and spelling errors present in the text and several sentences seem to miss a word.

Reviewer #3 (Remarks to the Author):

The authors report the development and application of the first JAZ subtype (i.e. JAZ9 and JAZ10) selective jasmonate agonist. Such compounds may represent important chemical tools for investigating the role of the individual JAZ proteins as well as may represent promising agrochemicals (after further chemical development). In addition, it is a very nice example for the use of structure-guided drug discovery in the field of protein-protein interaction stabilizers, a still challenging topic in chemical biology. It therefore addresses a highly relevant and timely research topic that in principle warrants publication in a prestigious journal such as *Nature Communications*.

The authors initiated their studies by generating a set of four coronatine analogues (differing in their stereochemistry at the coronamic acid and/or coronafacic acid building block). They then tested them for their potential to stabilize protein-protein interactions between the COI protein and different JAZ proteins or short model peptides representing the COI1-interacting Jas motifs via pulldown assays. These studies revealed that, as expected from previous studies, 'natural' coronatine is able to stabilize almost all COI1-JAZ interactions (and thus acts as a general stabilizer); one derivative denoted as ent6 however displayed a more selective mode-of-action.

To further enhance selectivity, the authors next used MD simulations (I cannot comment on their 'scientific validity' as this is out of my expertise) that indicated that ent6 may display a slightly different binding mode in COI-JAZ protein interactions including the presence of an additional binding pocket that may be targeted to enhance selectivity. The authors then use the Monte et al. approach (ref. 26) to 'fill' this potential binding pocket with oxime derivatives of ent6. One oxime derivative (Cpd 8 with a phenyl oxime moiety) turned out as a strong COI1-JAZ9 model peptide stabilizer (and to a lesser extent also of COI1-JAZ10 model peptide interactions), while the other interactions remained uninfluenced, thus indicating high selectivity.

Validation of these findings in planta was achieved by the use specific 35S::JAZ1/9/10:GUS reporter lines. Finally, the authors also showed that i) the JAZ9-selective compound 8 no longer impairs root growth but still triggers plant immunity (exemplified by plant infection assays with *Alternaria brassicola*) and ii) the influence of compound 8 (in contrast to mock and coronatine-treatment) on various jasmonate reporter and other genes, leading to a model for its mode-of-action.

Overall, this is a technically solid study that bases on previous findings that are however combined in a novel and original manner to solve a timely challenge in chemical biology, i.e. the development of JAZ-subtype specific jasmonate agonists. I therefore recommend its publication after some minor corrections.

I was particularly asked to check the chemical part of the manuscript:

The four coronatine isomers were synthesized and characterized in a previous study (ref. 29) and their preparation is thus not reported in this study. The synthesis of compounds 7-9 (oxime derivatives of coronatine) is however reported in this study, including a full chemical characterization.

I was however a little bit surprised by the low amounts of products that were generated, e.g. only 0.85 mg of compound 8 (major isomer) were obtained? Was this enough compound for all assays? In addition, I miss a description and characterization of the synthesized peptides. While I have no doubt that the authors have adequately made these peptides, for reasons of scientific adequacy, a synthesis protocol (peptide synthesis and oregon green functionalization) and a full characterization (i.e. a MS measurement and not only a LC trace as given in Fig. S2) should be added to the Supplementary Information.

Further points:

- 1) Line 45 – 48: this sentence does not sound well – please rephrase.
- 2) Line 77 and later sentences: I do not like the expression 'twisted 3D' structure. The authors have made a different stereoisomer (diastereomers or enantiomers) and the generated compounds are thus simply structurally close analogues – they are not twisted.
- 3) Line 81 and later sentences: '...providing the weak interactions...' – I do not think that the weighting 'weak' is right here and I would omit it. They stabilize the interaction that without the compound does not take place. This is not a weak interaction.
- 4) Line 88-91 – this is difficult to understand – please rephrase.
- 5) Line 108: '...and those of JAZ5 and 6 are almost identical..' – the sequence of the corresponding interacting motif given in Fig. S1 is identical, please rephrase.
- 6) Line 124: explanation for COI1-JAZ stabilizing properties of synthetic 2. One reason for the observed stabilization could be that indeed the 'wrong' major isomer is a weak stabilizer. An

alternative explanation however may be that 5% (i.e. the minor isomer) is a strong stabilizer and what can be seen is the effect from this compound. Could the authors comment on this?

7) Line 134: '...is over 100- to 1000-fold higher...' – I think it would be more accurate to state ent6 as only '..is over 100-fold higher..'

8) Line 153 and later sentences: 'vacancy' is an unusual wording in this context. I think it would be more appropriate to describe it as an 'additional unoccupied binding pocket'.

9) Line 200: for the jar1 mutant, a longer incubation time before analysis (8 h instead of 2 h) has been used. Could the authors comment on the reasons for this protocol change?

10) Line 576: the authors report the device for performing MALDI-TOF MS measurements – however, I did not find a use of this machine in the MS/SI?

11) The authors frequently use the word 'agonist' for the action of their coronatine analogues. While I think that this wording is correct for describing the physiological responses of their compounds (as coronatine is a hormone), I do not think that this wording should be used to describe the effect on the molecular protein-protein interaction (PPI) between COI1 and JAZ proteins. Here, the compounds act as 'PPI stabilizers' and not 'agonists'. I recommend to carefully check this wording in the whole manuscript.

12) Some Figures have not been linked with the main text, e.g. Fig. 3g, Fig. S6d and S6e. Please re-check accordingly.

13) Figure legend S2: There is no figure (d) (I think this is now figure 2d and 2e?). Please re-check and change accordingly.

Reviewer #4 (Remarks to the Author):

Jasmonate isoleucine (JA-Ile) is a critical plant hormone regulating plant defense responses against herbivores, pathogens, and other biotic and abiotic stresses. JA-Ile signals by promoting the interactions between the F-box protein COI1 and its substrate JAZ proteins, which result in JAZ ubiquitination and degradation. By destabilizing many members of the JAZ protein family, JA-Ile not only triggers plant defense responses but also affects other physiological functions, such as root growth and flower development. In this manuscript, the authors reported the development of a jasmonate-isoleucine mimetic compound that is highly selective to specific members of the JAZ protein family, thereby, achieving an up-regulatory effect on pathogen responses without compromising root growth. Overall speaking, the results are novel and the study has benefited from a multi-disciplinary approach, which includes chemical synthesis of multiple JA-Ile mimetics, biochemical analysis of compound-induced protein-protein interactions, structure-based docking and design, as well as functional studies in plants. After minor revisions, this reviewer recommends the publication of this manuscript in Nature Communication.

1. Page 16, line 314-315. "...this is the first example of a PPI agonist with high selectivity for specific protein subtypes" is an overstatement. The IMiD compounds are PPI agonists that can be highly selective for substrates within the same family. Nevertheless, it is appropriate to state that "...this is the first example of a JA-Ile agonist with high selectivity for specific protein subtypes."

2. Figure S2. There is no panel (d) despite its legend.

3. Figure 3(a)-(c) and Figure S5. Please consider remake these figures. This is unclear what the green/cyan blobs and the yellow sticks are. There is no label for the amino acids mentioned in the legend and the corresponding hydrogen bonds cannot be easily identified.

4. The authors used 0.5uM compound 8 for testing PPI interaction (Figure 3e), 1uM for testing root growth inhibition (Figure 4), 50uM compound 8 for triggering plant defense against the fungus A.

brassicicola (Figure 5), and 10uM for JA-responsive gene expression profiling (Figure 6). It will be informative and more rigorous if the authors can perform a dose response analysis of compound 8 in their pull down assay for two or three representative JAZ family members as they did for compound ent6. This will provide more confidence in correlating the results in Figure 4, 5, 6, with Figure 3.

Reviewer #5 (Remarks to the Author):

The Authors present a structure-based design of a JAZ subtype selective agonist able to decouple the plant defense response from growth-inhibition. Compound **8** is claimed to be a JAZ9/10-selective agonist for the COI1-JAZ protein-protein interaction. In a broader sense, and from a chemical point of view, computational and synthetic strategies aiming at the modification of a molecule to enhance a desired effect while switching off undesired side-effects is an area of active research and utmost importance.

The overall structure-based strategy designed by the Authors is logical. Five complexes of COI1 with JAZ subtypes (3, 9, 10, 11 and 12) and ligand COR were built by homology modeling starting from the COI1/COR/JAZ1 crystal structure. COR was replaced in those complexes with *ent***6** via molecular docking, and all the complexes subjected to molecular dynamics (MD) simulations with explicit water molecules. To this Reviewer, however, the *in silico* structure-based analysis leading to the design of molecule **8** cannot be properly understood from the content of the manuscript:

- a. Was the MD simulation of COI1/3/JAZ1 (PDB 3OMG) performed as a control/reference?
- b. Regarding the analysis of the hydrogen-bond pattern, and according to the crystal structure, it should be stressed that the oxygen of the ketone group of **3** and R496 do not form a hydrogen bond, since the distance O...HN(R496) is ~2.6 Å, with an unfavorable angle. To this end, the simulation mentioned in a. could be very helpful. Moreover, if hydrogen-bond patterns were to be analyzed, plots of distance vs. time should be very helpful, and could be included if necessary in Online Methods.
- c. It is not clear whether each system was run for 500 ns or 100 ns (totalizing 500 ns taken all together). In any case, it would be perhaps better to run 3-5 independent simulations for each system.
- d. The analysis of vacancies, for example, should be performed in a more quantitative way. On the contrary, only one snapshot is shown in Figs. 3 and S5 to exhibit vacancies. The role of water molecules should be also investigated.
- e. Taking into consideration comments a.-d., the rationale behind the design of compounds 7-9 should be explained in more detail.

Point-by-Point Responses to Reviewers' comments

Reviewer #1 (Remarks to the Author):

The development of compound **8** is interesting for potential agronomic applications and possible analysis of specific signaling networks governed by individual JAZs (in this case JAZ9 and JA10). Previous studies have also addressed the question of how to disengage the JA-mediated growth-defense trade-off (Campos et al 2016 Nat Commun) and engineered alternative COI1-JAZ ligands for biotech applications (Monte et al 2014 Nat Chem Biol), as well as ligands for the auxin receptor TIR1 that is structurally very similar to COI1. At the mechanistic level, this work is mostly an extension of the work done by Monte et al 2014 Nat Chem Biol. The manuscript is overall written and presented in a concise, clear and logical way.

However, several claims in this work are very weak and/or remain unsubstantiated:

1. A major claim in the paper is that treatments with compound **8** do not affect plant growth. This is supported by a single root length experiment with <20 individuals per genotype and a large variability (Fig4a,b). The authors should perform a proper dose-response growth analysis both in roots and aerial organs (eg. repetitive spraying) and compare the effects of compound **8** with effects elicited by COR. Importantly, those effects should also be tested in *jaz9/jaz10* mutants that should theoretically (if the authors' claims are correct and compound **8** is active only on COI1-JAZ9/10) result insensitive to compound **8** and still respond to COR or JA-Ile. Similarly, what are the long-term physiological consequences of compound **8** treatments?

Response: Thank you for this valuable suggestion. According to the reviewer's suggestion, we performed dose response growth analyses of COR (**3**)-derivatives for both roots and aerial organs (with conventional assay with agar plate or repetitive drop on the aerial parts). As shown in the revised Figures 4ab and S9, low concentration (0.1 μ M) of **3** induced growth inhibition for both organs, whereas the effects of **8** were diminished (0.1 μ M or 1 μ M) or significantly decreased even at the high concentration (10 μ M). Moreover, *jaz9/jaz10* mutants are insensitive to compound **8**, whereas still respond to **3** or *ent6*, as shown in the revised Figure S14. These results were added at lines 183-189 in page 10 as follows:

‘Growth inhibition and anthocyanin accumulation are well-known responses induced by jasmonates, including **3**,^{34,35} and it was hoped that these effects would not be observed for **8**. Growth analyses were undertaken using plates of ligand-containing agar or by repeatedly dropping the ligand solution on the leaf. It was found that the growth of both the root and aerial parts of *Arabidopsis* were strongly inhibited by **3**; less so by *ent6*; and hardly at all by **8** (Figures 4ab and S9ab).’

The growth analyses of *jaz9/10* mutants treated with **3**, *ent6* or **8** were also added at lines 230-233 in page 12 as follows:

‘Additionally, **8** did not affect the root growth in *jaz9/jaz10* mutants similar to WT plants, whereas all plants still responded to **3** or *ent6* (Figure S14), indicating that **8** is active only on COI1/JAZ9 *in planta*.’

2. No evidence is presented regarding potential off-target effects of compound **8**. How specific is the effect of this new ligand? Does this small molecule induce any other changes in addition to COI1-JAZ9/10 interactions? First, the specificity of **8** vs *ent6* on JAZ degradation (Fig.3f) should be extended to other JAZs (eg. JAZ3, JAZ11, and JAZ12) in a quantitative manner. Then, the authors should analyze the global differences in transcriptome changes occurring after compound **8** and COR treatments and evaluate the effects occurring in WT vs *jaz9/10* mutants. Furthermore, is compound **8** able to specifically induce COI1-JAZ binding but not TIR1-AUX/IAA binding?

Response: Thank you for your valuable comment to improve this manuscript. We analyzed the potential off-target effects of compound **8** with new JAZ-GUS lines (JAZ11-GUS and JAZ12-GUS in addition to the previous results with JAZ1-GUS, JAZ9-GUS and JAZ10-GUS). However, JAZ3-GUS cannot be used because we could not generate an appropriate 35S:JAZ3-GUS marker line. As shown in the revised Figures 3fg and S8, compound **8** selectively degraded only JAZ9/10-GUS lines whereas almost no degradation was observed in JAZ1/11/12-GUS lines. These results are newly added in lines 178-179 in page 10 as follows:

‘We further confirmed the JAZ-selectivity of *ent6* or **8** in an *in planta* assay using transgenic β -glucuronidase (GUS)-reporter *Arabidopsis* lines, including 35S:*JAZ1-GUS*, 35S:*JAZ9-GUS*, 35S:*JAZ10-GUS*, 35S:*JAZ11-GUS*, and 35S:*JAZ12-GUS* (Figure 3fg

and S8).²⁶ The degradation of the JAZ-GUS fusion protein was visualized as reduced GUS staining in all of five GUS-reporter *Arabidopsis* lines treated with **3**. In contrast, *ent6* and **8** triggered the degradation of JAZ-GUS in the JAZ9-GUS and JAZ10-GUS-reporter lines, whereas almost no degradation was observed in the JAZ1-GUS, JAZ11-GUS, and JAZ12-GUS line.’

Furthermore, we analyzed transcriptomic changes in *Arabidopsis* using WT treated with **3**, *ent6* or **8** and *jaz9/10* treated with **3** or **8** (Table S1). By the treatment of **8**, 99 genes were induced, of which 85 genes were induced both by **3** and *ent6* (Figure 4d, e). Among them, 40 genes (*ORA59*, *PDFs*, *ERF1* etc.) can be classified in JAZ9-regulated ones because their expression levels were lower in **8**-treated *jaz9* than in **8**-treated WT. Both **8** and **3** affect the expressions of other 59 genes in WT. This means that **8** functions as an analog of **3** and does not have additional off-target effects in WT. Moreover, the microarray result showed no up/down regulation of major IAA-responsive genes (*Aux/IAAs*, *ARFs* in Table S1; Mol Plant. 2008, 321-37. doi: 10.1093/mp/ssm021). Thus, the possible involvement of TIR1-AUX/IAA signaling can be excluded. Taken together, **8** affects the expression of JAZ9-related genes among the gene expression affected by **3**. These results were added in lines 191-196 in page 10-11 as follows:

‘To examine the mode of action (MOA) of **8**, DNA microarray analyses were carried out for comprehensive analyses of gene expression. When WT plants were treated with **8**, 99 genes were induced, 85 of which were also induced both by **3** and *ent6* (Figure 4d, e). Although several *JAZs* marker genes for early JA responses controlled by the COI1-JAZ co-receptor were included among these up-regulated genes, induction of these genes by **8** was lower than those by **3** and *ent6* (Figure 4f and Table S1).’

3. In addition to point 1 and 2, this work would be particularly strengthened if the authors could show that compound **8** is able to induce defense responses without compromising growth in other agronomically relevant plant species (eg. rice, wheat, maize, ..)

Response: We consider that application to other plant species will need further analyses that are beyond the scope of this manuscript.

MINOR POINTS

- Pg.4: Fig1b is cited before Fig1a and describes COR-dependent COI1-JAZ interaction. However, there is no explanation of COR in the text at this point

Response: '(Figure 1)' was deleted from Line 45.

- Line.53: The term 'player proteins' is unclear and unspecific, just as Line.61 'dream molecule'

Response: According to the review's comment, 'player proteins' in Line 53 was replaced by 'co-acting factors' at line 50 of revised manuscript, and 'dream molecule' in Line 61 was deleted and the whole sentence was rephrased (lines 54-59).

- Legend Fig2c: Specify what is 'Std'

Response: We replaced 'Std' into 'Input'.

- Line.163: why unexpectedly?

Response: According to the review's comment, we deleted the word, 'unexpectedly', in Line 163.

Reviewer #2 (Remarks to the Author):

This submission describes the synthetic design and functional analysis of a jasmonate agonist that may potentially uncouple the growth-defense trade-off that is normally encountered with the ‘natural’ jasmonate hormone JA-Ile. The authors attempted to create such a molecule by looking for coronatine derivatives with altered specificities to trigger JAZ-COII interaction. Whereas the authors have clearly developed a small molecule that has a lower affinity to 11 of the 13 JAZ proteins, the overall claims of the manuscript are not supported sufficiently by the data. Most importantly, the authors constantly use different concentrations of their molecules in the different assays, hence they do not show at all that the growth-defense trade-off is actually uncoupled. Secondly, one may wonder if the claimed JAZ-subtype-selectivity of their molecule would not rather reflect it being ‘only’ a weaker jasmonate molecule. Below my major comments are listed.

1. - The authors carry out all in planta assays at different concentrations, hence their data do not support the uncoupling of the growth-defense trade off yet. For compound **8** these are respectively 1 μM for the root growth inhibition (Fig 4a) and GUS degradation assays (Fig 3f) but 10 μM for the qRT-PCR analysis (e.g. Fig. 4d for the PDF1.2 expression induction) and even 50 μM for the fungal resistance assay. I wonder whether at a concentration higher than 1 μM of compound **8** no growth penalty would be observed, particularly given the fact that many ‘early’ JA-responsive genes, such as the JAZ, MYC2 and JA biosynthesis genes are clearly transcriptionally induced at 10 μM . To justify their major claim, the authors should therefore carry out all in planta assays at the 3 concentrations already used.

Response: Thank you for this valuable suggestion. According to the reviewer’s comments, we newly settled the concentration of the compound as 1 μM in all young planta assay (growth analyses, GUS degradation assays and qRT-PCR experiments). As shown in the revised Figure 4ab, **8** did not induce growth inhibition in either the roots or aerial parts of the plant. In contrast, the same concentration of the compound *ent6* induced JAZ9/10-GUS degradation and PDF1.2 expression compared to **3** (Figures 3f, 3g, and 4g). Moreover, we carried out the dose dependency experiments of this assay (Figure S9), and the results demonstrated that even at the highest concentration of **8** (10

μM) the effect on growth was much lower than that of **3**. These results clearly indicated that **8** selectively enhance defense responses without causing the growth inhibition at the same concentration.

2. - Along those lines, it will also be needed to test e.g. anthocyanin and/or glucosinolate accumulation in seedlings treated with compounds **3**, *ent6* and **8**, and see to what extent the activation of specialized metabolism is also triggered by e.g. compound **8**, which would further strengthen its value and utility. Likewise, additional ‘resistance’ assays such as wounding and/or bioassays with caterpillars after treatments with compounds **3**, *ent6* and **8**, should be carried out to assess the defense ‘range’ of the different compounds.

Response: Thank you for this valuable suggestion to improve this manuscript. According to the reviewer’s suggestion, we tested the anthocyanin accumulation in seedlings treated with **3**, *ent6* or **8**. As shown in Figure 4c, **3** strongly induced anthocyanin accumulation, whereas *ent6* or **8** weakly induced it and the effects were significantly lower than that of **3**. In contrast, we considered that wounding assay is not necessary because we cannot observe upregulation of *VSP2* gene by using our designed ligand **8** (Figure 4i). These results of anthocyanin accumulation were newly added at lines 183-190 in page 10 as follows;

‘Growth inhibition and anthocyanin accumulation are well-known responses induced by jasmonates, including **3**,^{34,35} and it was hoped that these effects would not be observed for **8**. Growth analyses were undertaken using plates of ligand-containing agar or by repeatedly dropping the ligand solution on the leaf. It was found that the growth of both the root and aerial parts of *Arabidopsis* were strongly inhibited by **3**; less so by *ent6*; and hardly at all by **8** (Figures 4ab and S9ab). Moreover, **3** strongly induced anthocyanin accumulation as previously reported,⁵ whereas *ent6* or **8** both only weakly induced it (Figure 4c).’

3. - Finally, the extent of an eventual uncoupling of growth and defense could be further strengthened by a genome-wide transcriptome analysis (by RNA-Seq). GO-enrichment term analysis could be carried out on such a data set. The experiment should include at least Col0 (WT) seedlings mock-treated and elicited with compounds **3**, *ent6*, and **8**.

Response: As suggested, we performed transcriptomic analysis using WT treated **3**, *ent6* or **8** and *jaz9/10* treated **3** or **8** (Figures 4d, 4e, S10, S17, and Table S1). The metabolism and stress overview (mapman analyses) added as Figure S17 very nice and basically addressed the GO-enrichment analysis. The results fully support previous conclusions. According to this change we added the sentence as follows in lines 302-304 of page 16:

‘Moreover, **8**-mediated transcriptomic changes corresponding to metabolism and regulation observed in WT and *jaz10* were dramatically suppressed in *jaz9* mutant (Figure S17).’

4. - The designed coronatine derivatives should be tested at a higher concentration range in the in vitro co-immunoprecipitation assays, in particular compounds *ent6* and **8**. For instance the experiment shown in Figure S4 should actually be carried out for all of the JAZ peptides and particularly for all those seemingly not being bound by *ent6*, namely JAZ1-4-5-6. Likewise, this experiment should also be conducted for compound **8** for all JAZ peptides. Only then, unambiguous conclusions can be drawn on whether there is true JAZ subtype-specificity or not. Along those lines, authors may have to edit some of their statements such as on page 8 lines 130-131 where the statement of strong and weak PPI induction by *ent6* for JAZ11-12 and JAZ9-10, respectively, does not seem to be supported by the data. I do not really see a difference in Fig. 2c. Likewise, to allow assessing the specificity of compounds 7-9 correctly, compound *ent6* should be included at 500nM in the assay shown in Fig 3e because now the authors compare only with the 100nM assay shown in fig 2.

Response: Thank you for this valuable suggestion. According to the reviewer’s suggestion, we re-assessed the selectivity of *ent6* and **8** for all combinations of COI1/JAZ co-receptors. As shown in revised Figure 3e, 500 nM of *ent6* (similar to compound **7-9**) selectively bound to COI1/JAZ3, 9-12, that is consistent with our previous conclusion. New dose response analyses of *ent6* with JAZ1 and 5/6, combined with the previous JAZ4 results indicated that the significantly lower affinity of *ent6* for these co-receptors (Figure S4b). Moreover, **8** can only induce PPI between COI1/JAZ9 and 10, whereas almost no band was observed in the new dose response of **8** for all combinations of COI1/JAZ (Figure S6). Overall, these results strongly supported our

previous claims in JAZ-subtype selectivity of our designed Coronatine derivatives. These results were newly added at lines 126-130 in page 8 as follows:

‘In contrast, *ent6* caused PPI for OG-JAZ1, 4 and 5/6 but only at concentrations of *ent6* as high as 3000 nM, a concentration over 100-fold higher than required for OG-JAZ3, 9, 10, 11, and 12 (Figure S4b), suggesting the significantly lower affinity of *ent6* for these co-receptors.’

5. - If all of the above analysis is carried out, it would be informative to map the selectivity of compounds *ent6* and **8** on a phylogenetic tree of the JAZ family, which will allow judging whether there is any phylogenetic correlation in their activity/specificity.

Response: Thank you for your valuable comment. According to the reviewer’s comments, we examined the correlation between the phylogenetic analyses of JAZ peptides and the selectivity of our designed ligands. As shown in Figure S4c, almost no correlation was observed, both in the case of *ent6* (strong affinity with JAZ3/11/12 and weak affinity with JAZ9/10) and **8** (JAZ9 and 10). These results were added to Figure S4c and lines 130-131 in page 8 as follows:

‘Little correlation was observed between the sequence homology of the JAZ subtypes and their individual affinity for *ent6* (Figure S4c).’

6. - The JAZ-GUS degradation experiment is very nice. To support the specificity of the PPI induction further, the authors could carry out a Y2H analysis for ligand dependent PPI between COI1 and different JAZs with the LexA system.

Response: As suggested, we examined COI1 interaction with all JAZs in presence of COR (**3**), *ent6* and **8** using the LexA and Gal4 systems. No interaction of any COI1-JAZ complex was induced by *ent6* nor **8**, although **3** induces some COI1-JAZ interactions. We think this can be attributed to the difference in the introduction amount of compounds because of their physicochemical nature. This is not an unexpected result since JA-Ile, the real natural ligand of COI1-JAZ, is only hardly positive at very high concentrations in this kind of tests. COR is more than 100-fold more active than JA-Ile. This is the reason why even at very high concentrations JA-Ile only induces a few COI1-JAZ interactions in Y2H. Therefore, if *ent6* or **8** are slightly less effective than

JA-Ile, it shouldn't be surprising to find negative results in this kind of assay. However, we consider that the result was confirmed by in vitro pull-down assay and in vivo JAZ-GUS assay. Pull-down is a much more sensitive technique.

7. - qRT-PCR analysis. It is not indicated which statistical analysis is conducted. Furthermore, in several cases the authors do not apply the statistical analysis correctly to allow supporting their conclusions. Most importantly, they should not only assess statistical significant differences between treatment with mock and **8** but also between treatment with **3** and **8** and with *ent6* and **8** (Figures 4 and S6-S7-S8). Only then they can draw conclusions about the specificity of **8**. Likewise they should also statistically compare the effects of **3** and **8** between Col-0 and the *jaz* mutants, and not only the effect of the compounds within the genotype (Fig. 6a-c).

Response: As the reviewer suggested, we newly examine the statistical analyses in qRT-PCR results to show the differences between treatment with not only mock and **3/ent6/8**, but also **3** and *ent6/8*, and with *ent6* and **8**, respectively. All of them strongly supported our previous conclusions. These results were newly added in the revised Fig. 4, 6, S9, S12, S13, S14, S15 and S16.

8. - The statement that ORA59 expression is more activated by **8** than by **3** or *ent6* is not true. This is actually also very clear from Fig6b! Likewise, the authors seem to cherry pick data for drawing conclusions. Whereas I agree that there is a differential trend for PDF1 expression, I fail to see a different trend between e.g. JAZ1-AOS-MYC2 on the one hand and ERF1-HEL-CHI on the other hand, contrary to the authors' view. These kinds of erroneous statements can be avoided if statistical analysis is properly conducted as suggested above. The overall trend of the expression data actually points to compound **8** being a 'weak' JA-like molecule rather than a specific one.

Response: Thank you for this valuable suggestion. As already mentioned, we fixed the concentration of the compounds (1 μ M) and reexamined the qRT-PCR experiments. The expression of ORA59, ERF1, HEL and CHI were significantly more activated by **8**

than **3** or *ent6*, which was confirmed with the new statistical analyses as shown in the revised Figure 4h and S12a. Moreover, the **8**-induced expressions of almost all other JA-responsive genes (*VSP1*, *MYC2*, *JAZ1*, *AOS*, *OPR3*, *LOX3* and *LOX2*) were significantly lower than **3**-induced ones (Figures 4i, 4j, 4f, S12b and S12c). These results support our findings that compound **8** can selectively enhance only the ORA59/ERF1-regulated defense-related genes, and that **8** is not merely a weak COR-analog.

9. - The expression data with the *ein2-1* mutant are also intriguing (Fig. S8). Also the effect of **3** on PDF1 and ORA59 expression is clearly down or abolished. I did not check whether this corresponds with data previously published on this mutant (if ever checked), but it would certainly be interesting to assess also the effects of **3** on the expression of other JA-response genes in this background. These data will be needed to further support the model proposed in Fig 6d-e. Eventually, this may possibly also help the authors to understand better the intriguing observation that in the *jaz9* mutant, nor in the *jaz10* or the quintuple *jaz* mutant, the expression of PDF1 is not released in the absence of JAs.

Response: Thank you for this valuable suggestion. Previous studies have revealed that jasmonates fail to induce *ERF1*, *ORA59* and *PDF1.2* in *ein2* background (*Plant Cell*, 2003, 15, 165-78; *Plant Cell*, 1996, 8, 2309-2323), which is consistent with our results with **3** and **8**. Additionally, as the reviewer suggested, we newly assessed the effects of our ligands on the expression of other JA-responsive genes in *ein2-1* mutant. As shown in the revised Figure S16d-f, **8**-triggered activation of MYC-branch genes (*VSP1* and *OPR3* as well as *MYC2*) was significantly enhanced in *ein2-1* background compared to WT plants. This activation can be attributed to the release from suppression by the ERF1/ORA59 branch in this mutant. These references and results were newly added at lines 239-242 in pages 13 as follows;

‘In contrast, activation of MYC-branch genes (*VSP1* and *OPR3* as well as *MYC2*) triggered by **8** was significantly enhanced in *ein2-1* background compared to WT plants (Figure S16d–f). This activation can be attributed to the release of the ERF1/ORA59 branch in this mutant.’

As the reviewer's suggestion, we newly assessed the effects of **3** or **8** on the expression of *PDF1.2/ORA59* in the quintuple jaz mutant (*jaz1/3/4/9/10*). As shown in the revised Figure S15, **8** does not induce upregulation of these genes in this mutant, whereas **3** still activates their expressions. These results also show that **8** selectively upregulates the expression of *ERF1/ORA59* branch through *COI1/JAZ9* *in planta*. These results were newly added at line 229-230 in page 12 as follows:

'The **8**-mediated upregulation of *PDF1.2/ORA59* expression was also impaired in *jaz1/3/4/9/10* quintuple mutants (Figure S15).⁴⁶,

Minor comments

- What does 'Std' in Fig 2c stand for?

Response: We replaced 'Std' into 'Input'.

- Figure 5, include the quantification of the lesion diameter after *Alternaria brassicicola* infection.

Response: We quantified the lesion diameter after *Alternaria brassicicola* add these new data in Figure 5c.

- Fig. S1. It would be helpful to the reader to indicate which JAZ peptides were eventually produced for the binding assays.

Response: We corrected Figure S1 and the legend as follows. 'Figure S1. Chemical structures of OG-derivatives as the epitope tags used in this study. Amino acid sequence alignment around the Jas motif of 13 *Arabidopsis thaliana* JAZs was coupled with X¹ (JAZ1-12) or X² (JAZ13). Acidic amino acids (D and E) are shown in red, and basic ones (K and R) are shown in blue.'

- Use of the English language should be verified. There are numerous grammar and spelling errors present in the text and several sentences seem to miss a word.

Response: The manuscript was checked by native English speaker.

Reviewer #3 (Remarks to the Author):

I therefore recommend its publication after some minor corrections.

1. I was particularly asked to check the chemical part of the manuscript:

The four coronatine isomers were synthesized and characterized in a previous study (ref. 29) and their preparation is thus not reported in this study. The synthesis of compounds 7-9 (oxime derivatives of coronatine) is however reported in this study, including a full chemical characterization.

I was however a little bit surprised by the low amounts of products that were generated, e.g. only 0.85 mg of compound 8 (major isomer) were obtained? Was this enough compound for all assays?

Response: Thank you for this valuable suggestion. For preparation of the revised manuscript and experiments, we resynthesized compound **8**. The details of the chemical synthesis protocol were revised in the revised supplementary information.

2. In addition, I miss a description and characterization of the synthesized peptides. While I have no doubt that the authors have adequately made these peptides, for reasons of scientific adequacy, a synthesis protocol (peptide synthesis and Oregon green functionalization) and a full characterization (i.e. a MS measurement and not only a LC trace as given in Fig. S2) should be added to the Supplementary Information.

Response: Thank you for your kind suggestion. According to the review's comment, we added MALDI-TOF MS data and spectra in 'Synthesis of epitope-conjugated JAZ peptides' in 'supporting method' of Supporting Information.

Further points:

1) Line 45 – 48: this sentence does not sound well – please rephrase.

Response: We rephrased the sentence to clarify the meanings as follows at lines 42-45 in page 4 in the revised manuscript): 'The upregulated JA responses depend on the degradation of specific subtype of JAZ repressor recruited by the COI1-JAZ co-receptor that in turn activates different subsets of transcription factors' into 'The upregulated JA responses depend on the degradation of the specific subtype of JAZ repressor recruited

by the COI1-associated ubiquitin ligase that in turn activates unique subsets of transcription factors’

2) Line 77 and later sentences: I do not like the expression 'twisted 3D' structure. The authors have made a different stereoisomer (diastereomers or enantiomers) and the generated compounds are thus simply structurally close analogues – they are not twisted.

Response: According to the reviewer’s comment, we replaced the word ‘twisted’ into ‘altered’ (lines 74 and 78 of page 5 in the revised manuscript). Additionally, ‘the partially distorted ligand’ in legend of Figure 1c was replaced by ‘the stereochemical isomer’.

3) Line 81 and later sentences: ‘...providing the weak interactions...’ – I do not think that the weighting 'weak' is right here and I would omit it. They stabilize the interaction that without the compound does not take place. This is not a weak interaction.

Response: We deleted the word ‘weak’ in Lines 81 and 83 according to the reviewer’s suggestion. And we rephrased whole sentence in lines 76-80 of page5 in revised manuscript.

4) Line 88-91 – this is difficult to understand – please rephrase.

Response: We rephrased the sentence as follows (lines 82-85 of page 6 in the revised manuscript): ‘Optically pure samples of the building blocks of **3**, (+)-coronamic acid (CMA, **4**),²⁹ and (+)-coronafacic acid (CFA, **5**),³⁰ were coupled to give naturally occurring **3**, the enantiomer *ent3*, and the stereochemical hybrid isomers, CFA-*ent*CMA (**6**) and *ent*CFA-CMA (*ent6*), respectively (Figure 1a).’

5) Line 108: ‘...and those of JAZ5 and 6 are almost identical..’ – the sequence of the corresponding interacting motif given in Fig. S1 is identical, please rephrase.

Response: We rephrased the description as follows (lines 101-102 of page 6 in the revised manuscript): ‘The Jas motifs of JAZ1 and 2 are almost identical and those of JAZ5 and 6 are identical (Figure S1a).’

6) Line 124: explanation for COI1-JAZ stabilizing properties of synthetic 2. One reason for the observed stabilization could be that indeed the 'wrong' major isomer is a weak stabilizer. An alternative explanation however may be that 5% (i.e. the minor isomer) is a

strong stabilizer and what can be seen is the effect from this compound. Could the authors comment on this?

Response: The latter explanation by the reviewer is correct. This is already confirmed in reference 5. To clarify this, we rephrased the sentence in Line 123-124 (lines 117-118 of page 7 in the revised manuscript) as follows: ‘, since the minor isomer (3*R*, 7*S*)-**2** is a strong agonist of the COI1-JAZ co-receptor’

7) Line 134: ‘...is over 100- to 1000-fold higher...’ – I think it would be more accurate to state ent6 as only ‘..is over 100-fold higher..’.

Response: According to the referee’s comment, we corrected the description as follows (line 128 of page 8 in the revised manuscript). ‘a concentration over 100-fold’

8) Line 153 and later sentences: ‘vacancy’ is an unusual wording in this context. I think it would be more appropriate to describe it as an ‘additional unoccupied binding pocket’.

Response: According to the referee’s comment, we replaced the word ‘a vacancy’ into ‘an additional unoccupied binding pocket’ in Line 153 (lines 157-158 of page 9 in the revised manuscript), into ‘additional unoccupied space’ in Line 154.

9) Line 200: for the jar1 mutant, a longer incubation time before analysis (8 h instead of 2 h) has been used. Could the authors comment on the reasons for this protocol change?

Response: The incubation time was settled for 8 h to observe the late-term JA-responsive genes (*PDF1.2* or *VSP1*), and for 2 h to observe other genes including the early-term JA-responsive genes (i.e. *ORA59/MYC2* or *AOS/OPR3/TAT3*). We replaced following description in ‘online methods’ of our manuscript (lines 695-702 of page 35 in the revised manuscript) to add the detailed conditions of these experiments.

‘Six day old plants were incubated in autoclaved 1/2 MS liquid medium containing each filtered compound under a 16 h light (118 $\mu\text{mol}\cdot\text{m}^{-2}\cdot\text{s}^{-1}$; cool-white fluorescent light)/8 h dark cycle at 21°C in a Biotron NC-220 growth chamber (Nippon Medical & Chemical Instruments Co., Ltd., Japan). The plants were treated with or without ligands (**3**, *ent6* or **8**, 0.1–10 μM) for 8 h (for late-term JA-responsive genes: *PDF1.2*, *VSP1*, *HEL*, *B-chi*, *LOX2*) or 2 h (for other genes such as transcription factors: *ORA59*, *MYC2*, *ERF1*, early-term JA-responsive genes: *JAZ1*, *AOS*, *OPR3*, *LOX3*, *TAT3*, and internal standard: *UBQ10*) in the same growth chamber. After treatments, total RNA’

10) Line 576: the authors report the device for performing MALDI-TOF MS measurements – however, I did not find a use of this machine in the MS/SI?

Response: Thank you for your kind observation. According to the reviewer's comment, we revised the peptide synthesis protocol and added MALDI-TOF MS data with spectra in Supporting Information.

11) The authors frequently use the word 'agonist' for the action of their coronatine analogues. While I think that this wording is correct for describing the physiological responses of their compounds (as coronatine is a hormone), I do not think that this wording should be used to describe the effect on the molecular protein-protein interaction (PPI) between COI1 and JAZ proteins. Here, the compounds act as 'PPI stabilizers' and not 'agonists'. I recommend to carefully check this wording in the whole manuscript.

Response: Thank you for this highly valuable suggestion. We corrected the descriptions as shown below:

a) line 60, p5

Old: 'a stereoisomer of **3** functions as a moderately selective JAZ-subtype agonist inducing COI1-JAZ co-receptor formation (Figure 1c).'

Revised: 'a stereoisomer of **3** functions as a PPI stabilizer with moderate JAZ-subtype selectivity inducing COI1-JAZ co-receptor formation (Figure 1c).'

b) lines 81-82, p5-6

Old: 'a subtype-selective COI1-JAZ agonist.'

Revised: 'a subtype-selective PPI stabilizer between COI1 and JAZ.'

c) lines 91-92, p6

Old: 'a JAZ subtype-selective agonist causing PPIs between COI1 and some of JAZ subtypes, including JAZ3.'

Revised: 'an agonist with JAZ subtype-selectivity causing PPIs between COI1 and some of JAZ subtypes, including JAZ3.'

d) lines 178-179, p10

Old: '**8** functions as a JAZ9/10-selective PPI agonist both *in vitro* and *in vivo*.'

Revised: '**8** functions as a JAZ9/10-selective PPI-stabilizer both *in vitro* and *in vivo*.'

e) lines 329-331, p17

Old: ‘In summary, we have developed a JAZ9/10-selective PPI agonist **8** for the COI1-JAZ co-receptor via rational design based on the *in silico* docking studies using a stereochemical isomer of **3**.’

Revised: ‘In summary, we have rationally developed an agonist **8** stabilizing JAZ9/10-selective PPI for the COI1-JAZ co-receptor via the *in silico* docking studies using a stereochemical isomer of **3**.’

12) Some Figures have not been linked with the main text, e.g. Fig. 3g, Fig. S6d and S6e. Please re-check accordingly.

Response: Thank you for this kind correction. According to the referee’s comment, we revised the main text and supplementary information as follows:

a) In line 173 of page 10 in the revised manuscript, we added Fig. 3g as ‘35S:JAZ12-GUS (Figure 3f, 3g, and S8)’.

b) We renumbered Figure S6 as Figure S12, and the data corresponding old “Figure S6d and e” (on CYP716A16, CYP94B1, and TAT3) were deleted from renumbered Figure S12.

13) Figure legend S2: There is no figure (d) (I think this is now figure 2d and 2e?). Please re-check and change accordingly.

Response: We deleted legend (d).

Reviewer #4 (Remarks to the Author):

After minor revisions, this reviewer recommends the publication of this manuscript in Nature Communication.

1. Page 16, line 314-315. "...this is the first example of a PPI agonist with high selectivity for specific protein subtypes" is an overstatement. The IMiD compounds are PPI agonists that can be highly selective for substrates within the same family. Nevertheless, it is appropriate to state that "...this is the first example of a JA-Ile agonist with high selectivity for specific protein subtypes."

Response: According to the reviewer's suggestion, we rephrased the sentence in lines 13-15 of page 2 and line 335-336 of page 17 as follows:

'To the best of our knowledge, this is the first example of a JA-Ile agonist with high selectivity for specific protein subtypes.'

2. Figure S2. There is no panel (d) despite its legend.

Response: We deleted legend (d).

3. Figure 3(a)-(c) and Figure S5. Please consider remake these figures. This is unclear what the green/cyan blobs and the yellow sticks are. There is no label for the amino acids mentioned in the legend and the corresponding hydrogen bonds cannot be easily identified.

Response: We revised Figures 3a-c and S5 to clarify the structure of complex and hydrogen bonds by using new software MOE (Chemical Computing Groups, Montreal, Canada). The details were added in 'General Materials and Methods' of 'Online Methods' in page 33 as 'The 3D structures of the ternary complex shown in Figure 3a-c and S5 were constructed using MOE 2016.08 software (Chemical Computing Groups, Montreal, Canada).'

4. The authors used 0.5uM compound 8 for testing PPI interaction (Figure 3e), 1uM for testing root growth inhibition (Figure 4), 50uM compound 8 for triggering plant defense against the fungus *A. brassicicola* (Figure 5), and 10uM for JA-responsive gene expression profiling (Figure 6). It will be informative and more rigorous if the authors can

perform a dose response analysis of compound 8 in their pull down assay for two or three representative JAZ family members as they did for compound ent6. This will provide more confidence in correlating the results in Figure 4, 5, 6, with Figure 3.

Response: According to the referee's comment, we newly performed dose response analyses of *ent6* and **8** in pull-down experiments for all combinations of COI1/JAZs, which clearly demonstrated the JAZ-subtype selectivity of these compounds. We added these results in Figures S4 and S6, respectively.

Reviewer #5 (Remarks to the Author):

The overall structure-based strategy designed by the Authors is logical. Five complexes of COI1 with JAZ subtypes (3, 9, 10, 11 and 12) and ligand COR were built by homology modeling starting from the COI1/COR/JAZ1 crystal structure. COR was replaced in those complexes with *ent6* via molecular docking, and all the complexes subjected to molecular dynamics (MD) simulations with explicit water molecules. To this Reviewer, however, the *in silico* structure-based analysis leading to the design of molecule **8** cannot be properly understood from the content of the manuscript:

- a. Was the MD simulation of COI1/3/JAZ1 (PDB 3OMG) performed as a control/reference?
- b. Regarding the analysis of the hydrogen-bond pattern, and according to the crystal structure, it should be stressed that the oxygen of the ketone group of **3** and R496 do not form a hydrogen bond, since the distance O...HN(R496) is ~2.6 Å, with an unfavorable angle. To this end, the simulation mentioned in a. could be very helpful. Moreover, if hydrogen-bond patterns were to be analyzed, plots of distance vs. time should be very helpful, and could be included if necessary in Online Methods.

Responses: Thank you for this valuable suggestion. We are grateful for these comments. We performed the MD simulations of the COI1/3/JAZ1 as a reference. The obtained typical bound structure of **3** was shown in the revised Figures 3a and S5a. These results clearly showed that the oxygen of ketone group of **3** and the NH1-proton of R496 (COI1) formed a hydrogen bond with proper bond distance and angle (the distance is 2.01 Å, and the angle is about 120°). Also, according to the referee's comment, we plotted the radial distribution functions (RDFs, $g(r)$) in Fig. S5a, to investigate the hydrogen bond networks between **3** and surrounding residues in the binding pocket. These RDFs were made by monitoring the distances between the oxygen atom of ketone of **3** and the hydrogen atoms of R496 (COI1) and A204 (JAZ1) for total 500 ns of MD time. The RDFs clearly showed that the NH1-proton of R496 (COI1) and NH-proton of A204 (JAZ1) mainly formed hydrogen bonds during the MD simulation, and these hydrogen bonds should key interactions for the binding of JAZ1. These details were newly added in page 34-35 as '*In silico* docking study' of 'Online Methods'.

c. It is not clear whether each system was run for 500 ns or 100 ns (totalizing 500 ns taken all together). In any case, it would be perhaps better to run 3-5 independent simulations for each system.

Responses: In this study, five independent 100 ns term MD simulations with different initial velocities were carried out for each system. (i.e. total 500 ns MD simulations were run for each system) We have confirmed that the obtained five results were very close. All simulation results (ex. RDF) were obtained from these statistically sufficient simulation data.

d. The analysis of vacancies, for example, should be performed in a more quantitative way. On the contrary, only one snapshot is shown in Figs. 3 and S5 to exhibit vacancies. The role of water molecules should be also investigated.

Responses: In this revision, we focused on the hydrogen bond networks of the compounds in the MD simulations of the COI1/3/JAZ1 and COI1/*ent6*/JAZ (JAZ3, JAZ9, JAZ10, JAZ11, JAZ12). The RDF analyses showed that the hydrogen bonds with the ketone oxygen of compounds should be important for the JAZ binding, and the hydrogen bonds with the ketone oxygen of *ent6* depend on the JAZ-subtype of systems (see figures 3 and S5). We think that these analyses and discussions should be more reasonable for design of compounds than the previous vacancies analysis using only one snapshot structure.

Also, since we have not observed any water molecules in the binding pocket of both crystal structure of COI1/3/JAZ1 and the snapshots obtained from the MD simulations of each model, the role of water molecules are not discussed in this study.

e. Taking into consideration comments a.-d., the rationale behind the design of compounds 7-9 should be explained in more detail.

Responses: Thank you for this valuable suggestion. As described above in the replies to the comments a-d, we have run long-term MD simulations of the COI1/3/JAZ1 and COI1/*ent6*/JAZ (JAZ3, 9, 10, 11 and 12) models and have observed the difference of

hydrogen bond networks around the ketone group of compounds in each system. These results suggest that the JAZ-subtype selectivity of *ent6* might be varied by these structural modifications of the ketone group. The related descriptions and figure captions are added in lines 142-160 of pages 8-9 as follows:

‘MD simulations were carried out for 500 ns to investigate the differences of hydrogen bond networks between the compounds and surrounding residues in the binding pocket of each system. In these MD simulations, the COI1/**3**/JAZ1 system was used as a reference, and the results of analysis were compared with those of the COI1/*ent6*/JAZ systems (JAZ3, 9, 10, 11 and 12). Figures 3a and S5a show the typical bound structure of **3** obtained from the MD simulation of COI1/**3**/JAZ1 and the radial distribution functions (RDF) curves for the possible hydrogen bond pairs. These results showed that the ketone oxygen of **3** mainly forms hydrogen bonds with the NH1-proton of R496 (COI1) and NH-proton of A204 (JAZ1) during the MD simulation, indicating the importance of these hydrogen bonds for the binding of JAZ1 with COI1. On the other hand, in cases of the COI1/*ent6*/JAZ systems, the formation of hydrogen bonds with the ketone oxygen of *ent6* depend on the JAZ-subtypes (see Figure 3bc and S5b-e). For instance, in the case of COI1/*ent6*/JAZ9, while the ketone oxygen of *ent6* formed a hydrogen bond with the NH1-proton of R496 (COI1), a less frequency of hydrogen bond formation was observed with the NH1-proton of A222 (JAZ9) (Figures 3b and S5e). These results suggest an additional unoccupied space around the ketone group of *ent6* would arise when complexed with COI1 and a part of JAZ subtypes, and thus the JAZ-subtype selectivity of *ent6*-derivatives might be altered by structural modifications of the ketone group.’

Moreover, we newly performed the *in silico* docking simulations of potential agonist **8** with COI1/JAZ9 and 10. The results clearly indicated that **8** bound to the same binding pocket of COI1/JAZ9 and 10, and the obtained binding poses were similar to that of **3** (Figure S7). Also, we have confirmed that these binding poses were kept during the long -term MD simulations. These details were newly added in the revised Online methods as ‘*In silico* docking study’ (p.34-35) and in lines 167-170 of pages 9-10 in the manuscript as follows:

‘The *in silico* docking simulations also showed that **8** can bind to the same binding pocket of COI1/JAZ9 and 10 (Figure S7). The obtained binding poses were similar to

that of **3** in the COI1/**3**/JAZ1 complex and were kept during the subsequent long-term MD simulation.’

Reviewers' comments:

Reviewer #1 (Remarks to the Author):

In this revised version, the authors have greatly improved the quality of the manuscript and constructively answered most of the reviewers' concerns. However, there are still major issues with this work, as well as some unaddressed responses to the reviewers.

A. Microarray Methods are incomplete: what was the age and tissue of plants used? How were the plants treated (concentration)? How many plants were used per biological replica? The standard cutoff for differentially expressed genes in microarray analyses is 2-fold change (sometimes 2.5), and not 10 as the authors have decided to do (legend in Fig. 4). Please use the standard cutoff (2-2.5FC) to state if a transcript is differentially expressed and list them in tables with appropriate statistics (i.e. p-value of the FDR, which is totally lacking from Table S1!). It would be helpful if authors could group the differentially expressed genes in GO groups to quickly determine if other pathways are affected by the treatments (i.e. off target effects). This was attempted in Fig. S17, but hormonal pathways (particularly ET) are left out. Finally, the raw data must be submitted to a public database (eg. GEO) where the reviewers can access it during the review process. Hence the accession number should be provided to reviewers.

B. Appropriate statistical tests must be used when comparing more than two samples, and sample size clearly indicated! I assume T-tests were used in many of the figures, but they are not appropriate here. Similarly $n=?$, is often missing. Please use for eg. an ANOVA followed by a Tukey-HSD in Fig. 3g; all graphs in Fig.4; Fig. 5b-c; Fig. 6a-c.

- Figure 3f-g. How many plants were tested? Line391 states 4 replicates, i.e 4 roots? This is an insufficient sample size: please use a standard sample size of 40-60 roots.

C. Lines 209-212. This is not a good argument and the presence of JA-Ile cannot be excluded based on jar1 data, because jar1 can still produce JA and JA-Ile (Suza and Staswick, 2008). Rephrase.

MINOR PINTS:

- text in introduction lines 25-26 and 28 is redundant and should be streamlined
- Fig 1d-e. It would be clearer to indicate COI1-GST and MBP-JAZ1/ MBP-JAZ3 in the immunoblot graphics, and then state which antibodies were used to assay them in the figure legend. As it stands now, it is not clear that JAZ1 is assayed in d, and JAZ3 in e, and the reader needs to refer to the text.
- line 106. According to FigS1, only JAZ13 has a C in its degron, and not a number of them.
- Add JAZ1 label to Fig. 2e
- legend for Fig.2e missing
- it is very hard to believe that plants shown in Fig. S9c are 6d old. Were they transferred or grown in the media? Please correct accordingly.

Additionally, I have been asked to check the answers to Reviewer #1 and #2, and done so as follows:

Reviewer #1

Response to point 1: dose-response growth analysis properly addressed. However, of course jas9 and 10 mutants will be insensitive to comp.8 at 1uM in root growth assays because the wt is already insensitive (Fig. S14)! The authors should repeat the experiment using 10uM of comp.8 at which concentration the WT is sensitive, but jaz9 and jaz10 should not be. And I think this is a critical figure that, if works as predicted, should be included in the main text.

Response to point 2: addressed but please cluster differentially expressed genes according to GO

terms for easier understanding of the data (see above point A).

Response to point 3: I still think this would be a simple experiment to perform (treat an agronomically important plant species with 8, assess growth and expression of defense marker genes) which would significantly boost the impact and relevance of this manuscript beyond Arabidopsis.

All minor points were addressed

Reviewer #2:

Response to point 1. I agree with Rev2 that this is an important point, which the authors mostly addressed, except for Fig.5 in which they used an unusually high dose (50uM) of comp.8. Given that the microarray of jaz9 and 10 treated with comp.8 still shows induction of many genes, it could be that comp.8 is just a weak agonist in planta so by increasing the dose to 50uM the authors see a COR-like defense response. In fact, at this dose the growth-defense effect is no longer uncoupled as already at 10uM comp.8 reduces growth. The authors should repeat the experiment in Fig.5 with 1uM and see if their hypothesis still stands. In addition, the difference (and limitations) between in planta assays and in vitro bindings should be clearly discussed in the discussion.

Response to point 2: only partly addressed, glucosinolates were not measured but secondary metabolism was extensively analysed in the microarray data.

Response to point 3: The suggested experiment was performed but GO analysis is missing (see my point A). Additionally, it is clear that both jaz9 and 10 are still sensitive to comp.8. (the transcriptome is very similar to COR treated plants, just weaker, and only a few genes are no longer differentially expressed). This means that JAZ9 and JAZ10 are probably not the sole targets of comp.8 and that there are other possible targets for comp.8 in planta. Perhaps single mutants are insufficient to fully address this question and the insensitivity (or not) of a jaz9 jaz10 double mutant is the only way to fully solve this question. Given that a double mutant was not analysed, the authors should phrase their conclusions more rigorously.

Response to point 4: Addressed

Response to point 5: Addressed

Response to point 6: Addressed and I agree with the authors regarding the pull-down.

Response to point 7: Statistics should be further improved (see my point B)

Response to point 8: Statistics should be further improved (see my point B)

Response to point 9: PDF1.2 is controlled synergistically by ET and JA. It is still possible that comp.8 is activating the ET pathway independently of JAZ9 or JAZ10 interaction with COI1. That is why a clear evaluation of off-target effects of comp.8 is so crucial. I have to say that at this point I am not fully convinced about the specificities of comp.8. Perhaps a quicker way to test this compared to the development of a jaz9/10 double mutant would be treatment of coi1-1 with comp.8 and assessment by qPCR of ET responsive genes like the ones found upregulated in your microarray (ERF1 and PDF1.2) as well as others that were found most strongly upregulated by comp.8 in jaz9 and jaz10 (eg. NATA1/ GDA1/ At1g73325/ or others). If they respond in coi1-1 you know there are offtarget effects of comp8 and then there is a clear difference btw the biochemistry and the in planta effects.

Minor comments: all addressed

Reviewer #3 (Remarks to the Author):

I am fine with the corrections made by the authors. This is a strong study and it should be published in Nat Commun.

Reviewer #4 (Remarks to the Author):

The authors have adequately addressed the concerns previously raised by this reviewer.

Reviewer #5 (Remarks to the Author):

The Authors have answered to my points in full.

Reviewer #6 (Remarks to the Author):

I was specifically asked to comment on the immunity-related aspects of this work. The authors show that the JA-Ile mimic 8 enhances the resistance of wt Arabidopsis plants to the necrotrophic pathogen *Alternaria brassicicola*. This finding is relevant because the same compound does not appear to affect plant growth parameters, thus uncoupling defense from growth phenotypes normally seen with Ja-Ile.

The 8-induced defense phenotype appears correlated with JA-associated gene expression changes. Also, gene expression responses of 8 in various jaz mutants associated 8-mediated responses with JAZ9. One note: it is difficult to review Figures 6a and c; asterisks indicate significant differences between invisible bars. Can you change the Y-axis to allow readers to see all bars (log2 rather than linear scales)?

As for the *Alternaria* experiment: from the pictures in figure 5a I am not convinced of the differences in lesion sizes that become apparent only after quantification in figure 5c. According to the Methods you used ImageJ; if you would measure the lesion area (rather than diameter), are the differences still as pronounced?

With the above in mind, I wonder if you can strengthen the immunity part of this work by adding gene expression data after infection. Do you see an exaggerated *A. brassicicola*-induced PDF1.2 transcript accumulation dependent on 8?

Alternatively or additionally, here is another thought fully admitting that I am not familiar with the jaz9 and jaz10 phenotypes in their interaction with pathogens. However: considering the dependence of 8-induced gene expression changes on JAZ9, do you observe 8-induced resistance to *A. brassicicola* in jaz9 and jaz10 (as a control) mutants?

Reviewers' comments:

Reviewer #1 (Remarks to the Author):

In this revised version, the authors have greatly improved the quality of the manuscript and constructively answered most of the reviewers' concerns. However, there are still major issues with this work, as well as some unaddressed responses to the reviewers.

A. Microarray Methods are incomplete: what was the age and tissue of plants used? How were the plants treated (concentration)? How many plants were used per biological replica? The standard cutoff for differentially expressed genes in microarray analyses is 2-fold change (sometimes 2.5), and not 10 as the authors have decided to do (legend in Fig. 4). Please use the standard cutoff (2-2.5FC) to state if a transcript is differentially expressed and list them in tables with appropriate statistics (i.e. p-value of the FDR, which is totally lacking from Table S1!). It would be helpful if authors could group the differentially expressed genes in GO groups to quickly determine if other pathways are affected by the treatments (i.e. off target effects). This was attempted in Fig. S17, but hormonal pathways (particularly ET) are left out. Finally, the raw data must be submitted to a public database (eg. GEO) where the reviewers can access it during the review process. Hence the accession number should be provided to reviewers.

Response: We have revised the materials and methods (pS29-S30) to provide the detailed information about plant age, number of plants per replication and treatment methods. Moreover, we have reanalyzed the microarray analysis and now the cut off value is 2.5-fold with FDR value <0.05 (Figure 4, S10 and Table S1 as a separate sheet). We have not only shown the general FDR value for each gene but also the FDR value (shown as p value) for individual combinations. Moreover, we have also performed the GO enrichment analysis which shows that **8** does not have any significant off target categories, as shown in the revised Figure 4f (according to this change, Figure 4f-j were renumbered as 4g-k). The results of GO enrichment analysis are presented in the same format as described by Campos et al. 2016 (DOI: 10.1038/ncomms12570). We hope that the revised version would be up to your standards (Figure 4f and Table S2 as a separate sheet). The microarray data has been deposited to GenBank with the accession number GSE110858.

B. Appropriate statistical tests must be used when comparing more than two samples, and sample size clearly indicated! I assume T-tests were used in many of the figures, but they are not appropriate here. Similarly $n=?$, is often missing. Please use for eg. an ANOVA followed by a Tukey-HSD in Fig. 3g; all graphs in Fig.4; Fig. 5b-c; Fig. 6a-c.

Response: Thank you for your valuable comments. We properly corrected the legend of all the figures to show the sample number, and significant differences were evaluated by one-way ANOVA/Tukey HSD post hoc test in Fig. 3g, Fig. 4, Fig. 5b-d, f-h, Fig. 6a-c in addition to the related Fig. S8, S9, S11, S12, S13, S14, S15, and S16.

- Figure 3f-g. How many plants were tested? Line391 states 4 replicates, i.e 4 roots? This is an insufficient sample size: please use a standard sample size of 40-60 roots.

Response: We corrected the legend of Figure 3g as follows:

‘(g) Quantification of GUS activity in 20 roots of 4-d-old 35S:JAZ1-GUS, 35S:JAZ9-GUS, and 35S:JAZ10-GUS plants ($n = 4$). Seedlings were pretreated as described above. Three independent replicates were measured, and values represent mean \pm s.d. (supplementary methods).’

According to this correction, we revised the corresponding Material and method in Supporting Information (pS28).

C. Lines 209-212. This is not a good argument and the presence of JA-Ile cannot be excluded based on *jar1* data, because *jar1* can still produce JA and JA-Ile (Suza and Staswick, 2008). Rephrase.

Response: Thank you for your supportive comment. We rephrased the sentence and added ref 46 (Suza and Staswick, 2008) in lines 209-213 of p11:

‘The induction of these genes by **8** would not be attributed to the presence of endogenously biosynthesized **2** because the same gene expression patterns for *PDF1.2*, *ORA59*, *VSP1*, and *MYC2* were also observed in the *jar1* mutant⁴⁴⁻⁴⁶ (in which the biosynthesis of **2** decreased) treated with *ent6* or **8** (Figure S13).’

MINOR PINTS:

- text in introduction lines 25-26 and 28 is redundant and should be streamlined

Response: We deleted the redundancy in line 28 of text and revised the text in line 27 of p3 as follows:

'JA-Ile (**2**) induces PPI between'

- Fig 1d-e. It would be clearer to indicate COI1-GST and MBP-JAZ1/ MBP-JAZ3 in the immunoblot graphics, and then state which antibodies were used to assay them in the figure legend. As it stands now, it is not clear that JAZ1 is assayed in d, and JAZ3 in e, and the reader needs to refer to the text.

Response: We revised the legend of Figures 1d-e and described the nature of antibodies as 'Goat' HRP-conjugated anti-GST antibody and 'Rat' anti-MBP antibody. Other details on these antibodies are described in SI. Additionally, we also revised the text in p6, lines 87-88) as follows:

'MBP-JAZ1 (Figure 1d), whereas three other isomers were inactive. In contrast, and to our surprise, both *ent6* and **3** caused PPI between GST-COI1 and MBP-JAZ3 (Figure 1e).'

- line 106. According to FigS1, only JAZ13 has a C in its degron, and not a number of them.

Response: We revised the sentences in lines 103-105 of p6-7 as follows:

'Thus, we prepared nine short peptides of 13 JAZ subtypes including JAZ1/2, JAZ3-6, JAZ9-12. And JAZ13 was also prepared as a negative control.'

- Add JAZ1 label to Fig. 2e

Response: Thank you for your kind suggestion. We corrected the Figure 2e.

- legend for Fig.2e missing

Response: I am sorry for your inconvenience. Figure legend of 2e is placed in lines 378-379 of p21 as follows: '(e) Pull-down assay of purified GST-COI1 (5 nM) with OG-conjugated JAZ1 peptide (10 nM) in the presence of **3**, *ent3*, **1** or **2** (100 nM);'

- it is very hard to believe that plants shown in Fig. S9c are 6d old. Were they transferred or grown in the media? Please correct accordingly.

Response:

Thank you very much for your kind correction. We corrected the legend of Figure S9c as follows:

‘(c, d) The effects of the repetitive treatment of the compounds (**3**, *ent6*, or **8** at 0.1–10 μ M, first at 6th day and second at 9th day) in the aerial part of WT *Arabidopsis* seedlings grown for 9 d on 1/2 MS medium (see supplementary methods).’

Additionally, I have been asked to check the answers to Reviewer #1 and #2, and done so as follows:

Reviewer #1

Response to point 1: dose-response growth analysis properly addressed. However, of course *jas9* and *jas10* mutants will be insensitive to comp.8 at 1 μ M in root growth assays because the wt is already insensitive (Fig. S14)! The authors should repeat the experiment using 10 μ M of comp.8 at which concentration the WT is sensitive, but *jas9* and *jas10* should not be. And I think this is a critical figure that, if it works as predicted, should be included in the main text.

Response:

Thank you for your kind suggestion. We newly added the similar results using 10 μ M of the compounds, as shown in the revised Figure S14c and d. In the results, **8**-triggered growth inhibition was observed but diminished in *jas9* mutant compared to wild-type plants or *jas10* mutant. These results combined with the previous experiments in *jas* mutants indicated that **8** is not specific, but mainly active on COI1/JAZ9 *in planta*. (see also response to point 9 from reviewer 2)

According to these results, we revised the following sentences in line 237-242 of p.12-13:

At higher concentration, **8** triggered growth inhibition, and this was suppressed in *jas9* mutant compared to WT or *jas10* (Figure S14cd). Thus growth inhibition by higher concentrations of **8** may be attributed to a weak effect of the high concentrations in other

COI1-JAZ co-receptors. However, these results demonstrate that **8** is mainly active on COI1/JAZ9 *in planta*.

Response to point 2: addressed but please cluster differentially expressed genes according to GO terms for easier understanding of the data (see above point A).

Response: We have also performed the GO enrichment analysis which shows that **8** does not have any significant off target categories. We hope that the revised version would be up to your standards. These results are presented in Figure S10 and Table S2.

Response to point 3: I still think this would be a simple experiment to perform (treat an agronomically important plant species with **8**, assess growth and expression of defense marker genes) which would significantly boost the impact and relevance of this manuscript beyond Arabidopsis.

Response:

Thank you for your suggestion. However, I am afraid to say that it will be examined in further studies including the mode of action by using COI1-JAZ combinations of crop plants.

Reviewer #2:

Response to point 1. I agree with Rev2 that this is an important point, which the authors mostly addressed, except for Fig.5 in which they used an unusually high dose (50uM) of comp.8. Given that the microarray of jaz9 and 10 treated with comp.8 still shows induction of many genes, it could be that comp.8 is just a weak agonist in planta so by increasing the dose to 50uM the authors see a COR-like defense response. In fact, at this dose the growth-defense effect is no longer uncoupled as already at 10uM comp.8 reduces growth. The authors should repeat the experiment in Fig.5 with 1uM and see if their hypothesis still stands. In addition, the difference (and limitations) between in planta assays and in vitro bindings should be clearly discussed in the discussion.

Response:

Thank you for your valuable comments. According to the reviewer's comment, we newly carried out assessment of defense responses and growth inhibition with repetitive administration of 50 µM compounds. The results were very similar to the ones in previous report by G. Howe *et al.* by using *PhyB* mutant (Figure 2c in *Nat. Commun.*, **2016**, 7, 12570). As shown in the revised Figure 5a-d, **8** did not induce growth inhibition, while it caused the expression of the PDF gene. In contrast, COR (**3**) caused both of growth inhibition and PDF expression in the same assay. These results clearly demonstrated that **8** uncouples the growth-defense tradeoff in higher dose with the aerial part of Arabidopsis plants. According to these results, we newly added Figure 5a-d (according to this change, Figure 5a-c were renumbered as 5e-e) and we revised the following sentences in line 215-226 of p 12 as follows:

Therefore, we assessed the effect of **8**, on plant defense against the fungus *Alternaria brassicicola* as well as on growth of the adult plants. As shown in Figure 5ab, repetitive addition of **3** induces strong growth inhibition on the aerial part of 5-week-old plants, whereas **8** did not. In contrast, both **3** and **8** activate the gene expression of *PDF1.2* or *ORA59* in the same way (Figure 5cd). Subsequently, treatment of the plant with COR (**3**) induced plant resistance against the fungal pathogen compared to the mock treatment (Figure 5e-g). Similarly, plants treated with **8** showed less chlorosis and harbored fewer fungal spores compared to the mock-treated plants after *A. brassicicola* infection which can be attributed to the upregulation of *PDF1.2* in the leaves (Figure 5e-g). These **8**-induced resistances were impaired in *jaz9*, whereas not affected in *jaz10* (Figure 5h). These results suggested that **8** can trigger comparable plant defense responses to **3**.

Response to point 2: only partly addressed, glucosinolates were not measured but secondary metabolism was extensively analysed in the microarray data.

Response:

Thank you for your kind suggestion. We newly analyzed the accumulation of glucosinolates. As shown in the revised Figure S9e, **3** strongly induced accumulation of glucosinolates, whereas *ent6* or **8** did not. According to this correction, we revised the corresponding Material and method in Supporting Information (pS28-29).

These results were newly added in lines 188-189 of p.10 as follows:

Moreover, **3** strongly induced anthocyanin or glucosinolate accumulation as previously reported,^{5,15} while *ent6* or **8** did not (Figure 4c and S9e).

Response to point 3: The suggested experiment was performed but GO analysis is missing (see my point A). Additionally, it is clear that both *jaz9* and *10* are still sensitive to comp.8. (the transcriptome is very similar to COR treated plants, just weaker, and only a few genes are no longer differentially expressed). This means that JAZ9 and JAZ10 are probably not the sole targets of comp.8 and that there are other possible targets for comp.8 in planta. Perhaps single mutants are insufficient to fully address this question and the insensitivity (or not) of a *jaz9 jaz10* double mutant is the only way to fully solve this question. Given that a double mutant was not analyzed, the authors should phrase their conclusions more rigorously.

Response:

Thank you for your valuable advice. According to reviewer's advice, we rephrased conclusion to reduce the strength by the following sentence in lines 237-242 of p13 to rephrase the conclusion:

At higher concentration, **8** triggered growth inhibition, and this was suppressed in *jaz9* mutant compared to WT or *jaz10* (Figure S14cd). Thus growth inhibition by higher concentrations of **8** may be attributed to a weak effect of the high concentrations in other COI1-JAZ co-receptors. However, these results demonstrate that **8** is mainly active on COI1/JAZ9 *in planta*.

Response to point 7: Statistics should be further improved (see my point B)

Response: Thank you for your valuable suggestion. We updated the statistics according to your suggestion.

Response to point 8: Statistics should be further improved (see my point B)

Response: Thank you for your valuable suggestion. We updated the statistics according to your suggestion.

Response to point 9: PDF1.2 is controlled synergistically by ET and JA. It is still possible that comp.8 is activating the ET pathway independently of JAZ9 or JAZ10 interaction with COI1. That is why a clear evaluation of off-target effects of comp.8 is so crucial. I have to say that at this point I am not fully convinced about the specificities of comp.8. Perhaps a quicker way to test this compared to the development of a jaz9/10 double mutant would be treatment of *coi1-1* with comp.8 and assessment by qPCR of ET responsive genes like the ones found upregulated in your microarray (ERF1 and PDF1.2) as well as others that were found most strongly upregulated by comp.8 in *jaz9* and *jaz10* (eg. NATA1/ GDA1/ At1g73325/ or others). If they respond in *coi1-1* you know there are offtarget effects of comp8 and then there is a clear difference btw the biochemistry and the in planta effects.

Response:

According to the reviewer's suggestion, we newly added the qPCR analyses in *coi1-1* mutant. As shown in the revised Figure S15cd, the **8**-mediated upregulation of *PDF1.2/ORA59* expression was impaired/suppressed in *coi1-1* mutant. Combined with the previous experiments in *jaz* mutants, these results indicated that **8** predominantly functions through COI1/JAZ9 *in planta*.

However, in the growth analyses with higher concentration of **8** (see also response to point 1), **8**-triggered growth inhibition was observed in *jaz9* mutants, indicating that other COI1-JAZ may affect the growth.

According to this change, we newly added Figure S15cd, ref48 (Xie *et al.* 1998) and revised the following sentences in lines 237-242 of p13:

At higher concentration, **8** triggered growth inhibition, and this was suppressed in *jaz9* mutant compared to WT or *jaz10* (Figure S14cd). Thus growth inhibition by higher concentrations of **8** may be attributed to a weak effect of the high concentrations in other COI1-JAZ co-receptors. However, these results demonstrate that **8** is mainly active on COI1/JAZ9 *in planta*.

Reviewer #6 (Remarks to the Author):

I was specifically asked to comment on the immunity-related aspects of this work. The authors show that the JA-Ile mimic **8** enhances the resistance of wt Arabidopsis plants to the necrotrophic pathogen *Alternaria brassicicola*. This finding is relevant because the same compound does not appear to affect plant growth parameters, thus uncoupling defense from growth phenotypes normally seen with Ja-Ile.

The **8**-induced defense phenotype appears correlated with JA-associated gene expression changes. Also, gene expression responses of **8** in various jaz mutants associated **8**-mediated responses with JAZ9. One note: it is difficult to review Figures 6a and c; asterisks indicate significant differences between invisible bars. Can you change the Y-axis to allow readers to see all bars (log2 rather than linear scales?)?

Response:

Thank you for your kind advice. We corrected the Y-axis of Figures 6a-c as log2-scale to avoid the 'invisible bars'.

As for the *Alternaria* experiment: from the pictures in figure 5a I am not convinced of the differences in lesion sizes that become apparent only after quantification in figure 5c. According to the Methods you used ImageJ; if you would measure the lesion area (rather than diameter), are the differences still as pronounced?

Response:

Sorry for the mistake in the figure legend, what we measured and is shown in figure 5g is actually the lesion area. According to this correction, we replaced the sentence in lines 222-226 of p12 as follows:

'Similarly, plants treated with **8** showed less chlorosis and harbored fewer fungal spores compared to the mock-treated plants after *A. brassicicola* infection which can be attributed to the upregulation of PDF1.2 in the leaves (Figure 5e-g). These **8**-induced resistances were impaired in *jaz9*, whereas not affected in *jaz10* (Figure 5h-i). These results suggested that **8** can trigger comparable plant defense responses to **3**.'

With the above in mind, I wonder if you can strengthen the immunity part of this work by adding gene expression data after infection. Do you see an exaggerated *A. brassicicola*-induced PDF1.2 transcript accumulation dependent on 8?

Response:

As suggested, we tested the *PDF1.2* and *HEL* gene expression in response to *Alternaria brassicicola* infection. In late infection stage (7 days), these senescence markers are induced in fungal-infected plants and their induction is reduced by treatment with 8 (see below). This result correlates with the 8-induced reduction of lesion area after fungal infection.

We also assess the fungal *AbB-TUB* expression; as expected, the treatment with 8 diminished the *Alternaria*'s gene expression (see below). This result is in agreement with the 8-induced decrease of fungal spores.

Analysis of gene expression by quantitative RT-PCR of adult WT *Arabidopsis* plants. Not infected plants (-) are compared to plants infected with *Alternaria brassicicola* treated with compound 8 (8) or without (Mock).

Alternatively or additionally, here is another thought fully admitting that I am not familiar with the jaz9 and jaz10 phenotypes in their interaction with pathogens. However: considering the dependence of 8-induced gene expression changes on JAZ9, do you observe 8-induced resistance to *A. brassicicola* in jaz9 and jaz10 (as a control) mutants?

Response:

We have analyzed resistance in the *jaz9* and *jaz10* mutants and results are consistent with our predictions and shown in new Figure 5h-i. According to this correction, replaced the sentence in p12, lines 222-226 as follows

‘Similarly, plants treated with **8** showed less chlorosis and harbored fewer fungal spores compared to the mock-treated plants after *A. brassicicola* infection which can be attributed to the upregulation of *PDF1.2* in the leaves (Figure 5c-g). These **8**-induced resistances were impaired in *jaz9*, whereas not affected in *jaz10* (Figure 5h). These results suggested that **8** can trigger comparable plant defense responses to **3**.’

REVIEWERS' COMMENTS:

Reviewer #1 (Remarks to the Author):

The manuscript has improved. Transcriptomics data are now properly analysed. The rational design of compound 8 is sound and in vitro pull-down data between COI1 and different JAZ proteins, as well as JAZ degradation assays in planta clearly show interaction specificities. All biochemical data are sound.

Nevertheless, the authors use low doses of compound 8 for their growth and transcriptomics analyses (0.1-10uM) and high doses for bioassays (50uM). With these discrepancies, they conclude that compound 8 induced defense genes without compromising growth. This is really not substantiated by their data, because 10uM of compound 8 reduced root and rosette growth! It rather seems that repetitive treatments with compound 8 do not alter rosette growth and that this type of treatment then renders plants more resistant to *Alternaria*. The authors should avoid unsubstantiated general statements, and rather describe accurately the outcomes of different treatments.

Minor points:

1. Fig S9c-d could be included somewhere in Fig 4, showing that repetitive treatment does not alter growth. It seems that repetitive treatment with compound 8 does not compromise rosette growth, while growing plants continuously in the media does. This should be clearly stated and discussed.

Line 279. Compound 8 DOES affect root growth (Fig S9)!! Rephrase throughout the manuscript that repetitive treatment does not alter growth, as root and rosette growth is clearly inhibited when plants are grown continuously in the media.

2. Following multiple comparisons tests, it is recommended to show letters on top of bars instead of stars and lines. As presented now, the data is very confusing and ambiguous

Eg. Fig1f

JAZ1: mock = a; 3 = b; ent6 = a or b?; 8 = a (or whatever is the outcome of the statistics test)

JAZ9: mock = a; 3, ent6, 8 = all b (I guess..)

See fig2c here as an example: <https://www.nature.com/articles/s41598-018-27904-1>

This confusion applies to all figures in the manuscript and should be corrected.

Reviewer #6 (Remarks to the Author):

Thanks for your elaborate answer and revision.

REVIEWERS' COMMENTS:

Reviewer #1 (Remarks to the Author):

The manuscript has improved. Transcriptomics data are now properly analysed. The rational design of compound 8 is sound and in vitro pull-down data between COI1 and different JAZ proteins, as well as JAZ degradation assays in planta clearly show interaction specificities. All biochemical data are sound.

Nevertheless, the authors use low doses of compound 8 for their growth and transcriptomics analyses (0.1-10uM) and high doses for bioassays (50uM). With these discrepancies, they conclude that compound 8 induced defense genes without compromising growth. This is really not substantiated by their data, because 10uM of compound 8 reduced root and rosette growth! It rather seems that repetitive treatments with compound 8 do not alter rosette growth and that this type of treatment then renders plants more resistant to *Alternaria*. The authors should avoid unsubstantiated general statements, and rather describe accurately the outcomes of different treatments.

Response: Our unclear description about the growth stage of plant used in these experiment may cause difficulty for understanding. We used plant at appropriate growth stage treated with standard concentration of sample for each biological evaluation. We have corrected and described the plant growth stage (i.e. seedlings or adult plants) treated at 0.1-10/50 uM in Figure 4/5 clearly to distinguish each experiment (Lines 181, 281, 285, 914, 917, 920, 921). We also replaced from “the root growth” to “the growth” to include root growth as well as the one of aerial part (Line 281).

Minor points:

1. Fig S9c-d could be included somewhere in Fig 4, showing that repetitive treatment does not alter growth. It seems that repetitive treatment with compound 8 does not compromise rosette growth, while growing plants continuously in the media does. This should be clearly stated and discussed.

Line 279. Compound 8 DOES affect root growth (Fig S9)!! Rephrase throughout the manuscript that repetitive treatment does not alter growth, as root and rosette growth is clearly inhibited when plants are grown continuously in the media.

Response: Thank you for the valuable comments. We newly added the description of the data in Supplementary Figure 9cd (Line 188-190) as follows: Almost no growth inhibition was observed with the repetitive treatment of both *ent6* and **8** in the aerial part of *Arabidopsis* (Supplementary Figure 9cd).

2. Following multiple comparisons tests, it is recommended to show letters on top of bars instead of stars and lines. As presented now, the data is very confusing and ambiguous

Eg. Fig1f

JAZ1: mock = a; 3 = b; ent6 = a or b?; 8 = a (or whatever is the outcome of the statistics test)

JAZ9: mock = a; 3, ent6, 8 = all b (I guess..)

See fig2c here as an example: <https://www.nature.com/articles/s41598-018-27904-1>

This confusion applies to all figures in the manuscript and should be corrected.

Response: According to this suggestion, we showed letters on top of bars instead of stars and lines in all figures (Fig. 3, 4, 5, 6, Supplementary Fig. 8, 9, 11, 12, 13, 14, 15, 16).

Reviewer #6 (Remarks to the Author):

Thanks for your elaborate answer and revision.